# Tracking the movement of discrete gating charges in a voltage-gated potassium channel

Michael F Priest[1†], Elizabeth EL Lee[1†], Francisco Bezanilla[1,2]*

[1]Committee on Neurobiology and Department of Biochemistry and Molecular Biology, University of Chicago, Chicago, United States; [2]Centro Interdisciplinario de Neurociencia de Valparaíso, Facultad de Ciencias, Universidad de Valparaíso, Valparaíso, United States

**Abstract** Positively charged amino acids respond to membrane potential changes to drive voltage sensor movement in voltage-gated ion channels, but determining the displacements of voltage sensor gating charges has proven difficult. We optically tracked the movement of the two most extracellular charged residues (R1 and R2) in the Shaker potassium channel voltage sensor using a fluorescent positively charged bimane derivative (qBBr) that is strongly quenched by tryptophan. By individually mutating residues to tryptophan within the putative pathway of gating charges, we observed that the charge motion during activation is a rotation and a tilted translation that differs between R1 and R2. Tryptophan-induced quenching of qBBr also indicates that a crucial residue of the hydrophobic plug is linked to the Cole–Moore shift through its interaction with R1. Finally, we show that this approach extends to additional voltage-sensing membrane proteins using the *Ciona intestinalis* voltage-sensitive phosphatase (CiVSP).

**\*For correspondence:**
fbezanilla@uchicago.edu

[†]These authors contributed equally to this work

## Editor's evaluation

How the voltage sensor moves to produce voltage-dependent opening of ion channels is still an interesting and open question. The authors measured the presumed movement of the voltage sensor of the Shaker potassium channel using the fluorescence quenching of bimane by tryptophan, a method previously used to measure rearrangements in other proteins. To measure the trajectory of the first two voltage sensor charged residues (R1 and R2), these authors modified R1 and R2 cystine mutant channels with a derivative of bimane that has a positive charge similar to the native arginines. They then mutated residues around the sensor to tryptophan and measured the state-dependent quenching with voltage steps. They observed a pattern for the state-dependent quenching that suggests that the charge pathway during activation is a rotation and tilted translation. Furthermore, they suggest that this pathway for activation is distinct from the pathway for deactivation. Finally, they show initial evidence that this approach could also be applied to the voltage-sensitive phosphatase CiVSP.

## Introduction

The nature of the motion of the voltage sensor in voltage-gated ion channels has been a subject of intensive research. This motion is driven by voltage changes sensed by positively charged amino acids (typically arginines) found on the fourth transmembrane segment (S4) of each monomer of the tetrameric channel. The number of charged amino acids driving this motion varies from channel to channel, but has been shown in the canonical voltage-gated Shaker potassium channel (Kv) to consist of the

**Figure 1.** The basic principles of qBBr gating charge tracking in time. (**A**) Chemical structure of monobromo(trimethylammonio)bimane, a small positively charged (blue) fluorophore with the ability to conjugate to cysteine through a bromide group (yellow). (**B**) Size comparison of qBBr-Cys (top) to an arginine (bottom) (blue, nitrogen; red, oxygen; sulfur, yellow; gray, carbon). (**C**) Cartoon schematic of qBBr-tryptophan distance-based quenching with representative fluorescence data for unquenching (top), no effect (center), and quenching (bottom).

The online version of this article includes the following figure supplement(s) for figure 1:

**Figure supplement 1.** Single molecule fluorescence trace simulations in comparison to macroscopic fluorescence traces.

four most extracellular arginines (*Aggarwal and MacKinnon, 1996*; *Seoh et al., 1996*). However, research into the motion of these individual gating charges has suffered from experimental limitations. Replacement of arginine by histidine together with pH titration allows the study of gating charge end positions during activation, but does not show the charge's actual trajectory (*Starace and Bezanilla, 2001*; *Starace et al., 1997*). Cross-linking studies replace residues with cysteines to see which residues interact with each other in different conformational states, necessarily abrogating the positive charge of any gating charge if they are examined directly (*Broomand et al., 2003*; *Henrion et al., 2012*; *Lainé et al., 2003*), and accessibility studies have the same limitation (*Larsson et al., 1996*; *Yang and Horn, 1995*). Similarly, site-directed fluorimetric approaches typically replace a residue with a cysteine and then attach a fluorescent dye, providing the additional advantage of being able to monitor conformational changes in real time, but do not directly follow the movement of the gating charges (*Cha and Bezanilla, 1997*; *Mannuzzu et al., 1996*; *Priest and Bezanilla, 2015*). Positively charged adducts such as methanethiosulfonate-ethyltrimethylammonium linked to cysteines placed at gating charges can be used to replace the positive charge and characterize discrete gating charges (*Ahern and Horn, 2004*; *Ahern and Horn, 2005*; *Baker et al., 1998*; *Larsson et al., 1996*). However, these replacement charges cannot be rapidly monitored, limiting their use for observing conformational changes.

Ideally, one would like to follow the movement of individual gating charges in real time as they respond to changes in the electric field. This requires a fluorophore that is comparable to the gating arginines in size and charge. We used monobromo(trimethylammonio)bimane (qBBr), a small molecule fluorescent dye with a permanent positive charge (*Figure 1A*). While bulkier than an arginine (MW≈ 295 versus 101) (*Figure 1B*), modeling the conjugation of qBBr to a cysteine substituted into the two most extracellular gating charges in a Shaker-type voltage-gated potassium channel (R362 or R1 and R365 or R2) (*Chen et al., 2010*; *Long et al., 2005*; *Pettersen et al., 2004*) provided a distance between the carbon backbone and the positive charge of qBBr of ~5.5 Å, very close to the analogous distance of ~5.3–6.5 Å for arginine. Therefore, qBBr attached to a cysteine at the endogenous position of the gating charge may mimic the positive gating charge. In addition to being positively charged, qBBr has useful fluorescence quenching properties. Tryptophan has been shown to strongly quench qBBr fluorescence, with weak quenching by tyrosine, and no quenching from various other amino acids including histidine, phenylalanine, methionine, aspartate, or arginine (*Mansoor et al.,*

2010). This phenomenon of 'tryptophan-induced quenching' in bimane dyes generally (*Mansoor et al., 2002*) has been taken advantage of to measure conformational rearrangements of various membrane proteins, including the β2-adrenergic G-protein-coupled receptor (*Yao et al., 2006*), a cyclic nucleotide-gated ion channel (*Islas and Zagotta, 2006*), the BK channel (*Semenova et al., 2009*), a proton-gated ion channel (*Menny et al., 2017*), and a lactose permease (*Smirnova et al., 2014*). In general, these studies use the fluorescent quenching produced by an interaction between bimane and a particular tryptophan, whether native or engineered into the protein, to provide insight into how the protein moves during conformational transitions.

Here, we perform site-directed voltage-clamp fluorimetry on qBBr-bound Shaker Kv channels expressed in *Xenopus laevis* oocytes. By substituting a cysteine for a native gating charge and then covalently attaching qBBr to this site, we produce a fluorescent mimic of a discrete charge in the voltage sensor of the channel. Individual native or engineered tryptophans in the channel are then used to quench the fluorescence of the charged qBBr as it moves in response to changes in membrane potential. We used this system to investigate the pathway taken by the individual gating charges R1 and R2 during activation and deactivation. In general, the movement of R1C- and R2C-qBBr follow a tilted translation across the membrane and rotation, giving new information on the motion of the voltage sensor that had been inferred from recent consensus models (*Henrion et al., 2012*; *Li et al., 2014*; *Vargas et al., 2011*; *Vargas et al., 2012*). Of particular interest, during channel activation from physiological resting membrane potentials R1C-qBBr does not appear to interact with F290W, the most intracellular residue of the hydrophobic plug (*Chen et al., 2010*; *Lacroix and Bezanilla, 2011a*), also called the gating charge transfer center (*Tao et al., 2010*). Instead, R1 only interacts with F290W at extremely negative potentials, suggesting that R1 does not normally move past the hydrophobic plug and provides the basis of the Cole–Moore shift (*Cole and Moore, 1960*). The Cole–Moore shift is a widespread phenomenon in voltage-gated ion channels: severe hyperpolarizations of the membrane result in lags in ionic, as well as gating, currents (*Cole and Moore, 1960*; *Hoshi and Armstrong, 2015*; *Stefani et al., 1994*; *Taylor and Bezanilla, 1983*). Finally, the qBBr-tagging technique described here should also be transferrable to other voltage-sensing membrane proteins; as a proof-of-principle, we demonstrate its use in the voltage-sensitive phosphatase CiVSP (*Murata et al., 2005*).

## Results

### The basic principles of qBBr gating charge tracking in time

The idea of the present approach is to study the translocation of the gating charge (now a fluorophore, qBBr) as the membrane potential is changed using the specific quenching of qBBr by a tryptophan (W) that is positioned nearby or in the path of qBBr.

If we were tracking fluorescence at the single molecule level, the fluorescence signal we would observe would depend on the position of the W with respect to the moving qBBr (*Figure 1—figure supplement 1*). Let us assume that we are applying a positive voltage to activate the voltage sensor that moves between two discrete positions. We can distinguish three extreme cases schematically (*Figure 1C*, left panel). If the W is near the resting position of the qBBr, we would see a sudden increase in fluorescence when the qBBr moves away from it and the time lag before that increase corresponds to the waiting time of the sensor before it jumps across the energy barrier (*Figure 1C*, left, top trace). On the other hand, if the W is in the path of qBBr (*Figure 1C*, left, middle), we would see an extremely brief decrease of fluorescence as the qBBr passes by the quenching group. Finally, if the W is near the final position of the qBBr (*Figure 1C*, left, bottom) we would see a sudden decrease in fluorescence that would be maintained when the qBBr reaches that point. The duration of the high fluorescence period is the waiting time before the sensor crosses the energy barrier.

However, since we are looking at a large ensemble of molecules the macroscopic fluorescence signals we see are the result of an ensemble of voltage-sensing domains (VSDs) moving, each one at a different time according to its first latency duration, generating a continuous change in fluorescence. The fast event for the case of a W in the middle of the qBBr path would not be visible in the ensemble because the crossing of the VSDs is unsynchronized and if it is exactly in the middle no signal will be seen (*Figure 1C*, right panel, middle trace). The other two extreme cases will generate a continuous

quenching or unquenching depending on the placement of the tryptophan (*Figure 1C*, right, top and bottom).

A model of quenching of the fluorescence as a function of voltage and time of the moving qBBr in the voltage sensor can be proposed based on the known quenching properties of W on qBBr. As the concentration (*C*) dependence of W on qBBr follows the classical Stern–Volmer equation (*Islas and Zagotta, 2006*) we can rewrite the Stern–Volmer equation in terms of distance *d* using the relation $d \sim 1/C^{1/3}$ as follows

$$F\left(t\right) = F_o \, \frac{1}{1+\left(\frac{\lambda}{d}\right)^3} \tag{1}$$

where *F(t)* is qBBr fluorescence, $F_0$ is the maximum qBBr fluorescence (no quenching), *d* is the distance from qBBr to W and $\lambda$ is the distance to quench half of the fluorescence. Now, we assume that qBBr moves in a trajectory represented by *x* between the initial position, $x_0$, and the final position, $x_f$, and *W* is located somewhere near the *x* trajectory, $x_W$. Let us consider first that qBBr moves between two states and define $P_0(t)$ and $P_f(t)$ as the probabilities that qBBr is in $x_0$ and $x_f$, respectively. Then, $P_0(t) + P_f(t) = 1$ and we can write the time course of the fluorescence as

$$\frac{F(t)}{F_0} = P_0\left(t\right)\left[\frac{1}{1+\left(\frac{\lambda}{|x_0-x_w|}\right)^3} - \frac{1}{1+\left(\frac{\lambda}{|x_f-x_w|}\right)^3}\right] + \frac{1}{1+\left(\frac{\lambda}{|x_f-x_w|}\right)^3} \tag{2}$$

This means that the time course of fluorescence will have the time course of $P_0(t)$ and will not depend on the position of *W* in the trajectory.

If we now assume that the number of states is more than 2, say *i* = 1, 2, 3, …, *n*, we can write a general solution for the fluorescence

$$\frac{F(t)}{F_0} = \sum_{i=1}^{n} P_i\left(t\right)\left[\frac{1}{1+\left(\frac{\lambda}{|x_i-x_w|}\right)^3}\right] \tag{3}$$

$P_i(t)$ will have *n* − 1 eigenvalues, so the time course of fluorescence will now depend on the position of the *W* in the trajectory, because $P_i(t)$ will be weighted by the exponential term that includes the distance to each position. Simulations with three states show fluorescence time courses with more than one exponential component and, depending on the position and dwell time in the intermediate state, the response may be biphasic (i.e., fluorescence increases and decreases) during the depolarization. This is expected because as it traverses its trajectory the qBBr will go through periods when it will be closer to the *W* and others when it will be further away. This is a prediction that will be important in interpreting the experimental results.

## qBBr mimics a native gating charge

While qBBr crucially contains a permanent charge (*Figure 2A*) like the arginines that constitute the gating charges in the Shaker Kv channel, this is not a guarantee that the voltage sensor will continue to behave normally with a fluorescent dye substituted for a gating charge. To test whether this was the case, we first separately mutated R1 and R2 in a nonconducting (W434F) (*Perozo et al., 1993*), nonfast inactivating (Δ6–46) (*Hoshi et al., 1990*) Shaker channel to cysteines. Following expression of these constructs in *X. laevis* oocytes, qBBr was conjugated to either of these charges by incubation of the oocytes in a depolarizing solution containing qBBr. Oocytes containing qBBr conjugated to R1C or R2C were then simultaneously recorded optically and electrophysiologically using the cut-open oocyte voltage-clamp technique. In both cases, the observed gating currents may be a combination of voltage sensors that have been labeled with qBBr and unlabeled voltage sensors that retain a bare cysteine. This means the gating currents may be from a combination of R1C and R1C-qBBr subunits (or R2C- and R2C-qBBr subunits), potentially producing underestimates of the functional effects of qBBr labeling on gating currents (*Figure 2B*). For R1C-qBBr, gating currents appeared to be faster than those of the wild-type channel but still comparable in voltage dependence (*Figure 2A, B*). Consistent with previous results (*Larsson et al., 1996*; *Papazian et al., 1991*), the charge versus voltage (*QV*) relationship for R2C is shallow and has two components but when qBBr is conjugated (R2C-qBBr), it becomes less shallow and shows only one component. Still, R2C-qBBr does not recover the voltage dependence of the WT channel indicating that only a fraction of R2C channels have been

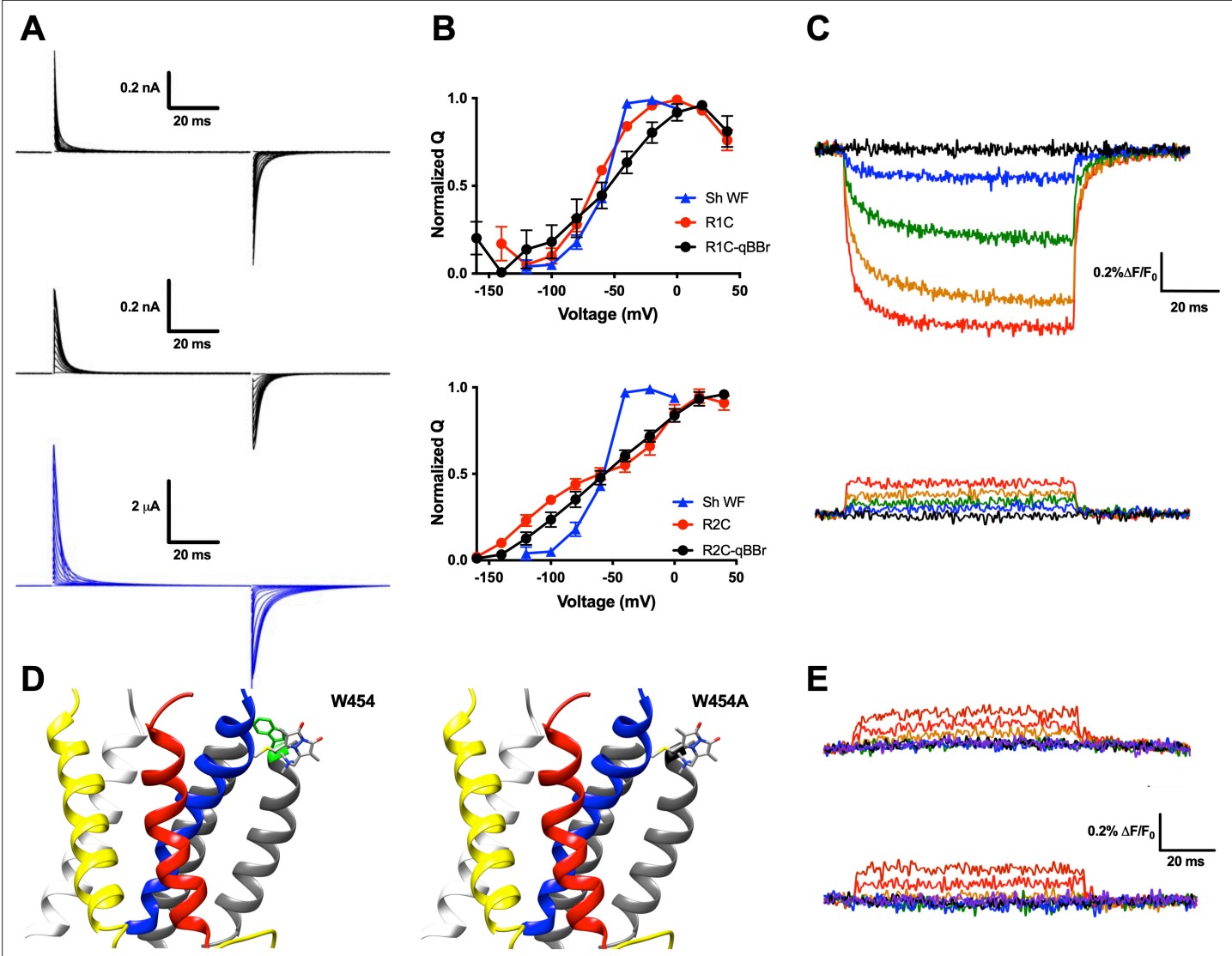

**Figure 2.** qBBr mimics a native gating charge. (**A**) Representative gating current traces of Shaker R1C-qBBr (top), R2C-qBBr (middle), and Shaker W434F (bottom). (**B**) Normalized charge (Q) versus voltage (V) QV curves of R1C (top, red circles, $n = 4$), R1C-qBBr (top, black circles, n = 4), R2C (bottom, red circles, $n = 5$), and R2C-qBBr (bottom, black circles, $n = 3$), compared to Shaker W434F (blue triangles both, $n = 3$). Data are shown as mean ± standard error of the mean (SEM). (**C**) Representative fluorescence traces of R1C-qBBr (top) and R2C-qBBr (bottom). In all figures, membrane potentials during the pulse are: brown, + 80 mV; red, +40 mV; orange, 0 mV; green, −40 mV; blue, −80 mV; black, −120 mV; purple, −160 mV. (**D**) Structure of an active state voltage sensor showing R1C-qBBr (stick model) with the position of the in silico mutation 454 from a tryptophan (left, green) to an alanine (right) (PDB: 3LUT) (**Chen et al., 2010**). In all Shaker structures, transmembrane domains (S1–S6) are colored: S1, white; S2, yellow; S3, red; S4, blue; S5 and S6, gray. (**E**) Representative fluorescence traces for R1C-qBBr:W454A (top) and R1C-qBBr:W454F (bottom).

The online version of this article includes the following source data and figure supplement(s) for figure 2:

**Source data 1.** Source data for normalized QVs in *Figure 2B*.

**Figure supplement 1.** R1C-qBBr:W454A and R2C-qBBr fluorescence signals are not produced by endogenous tryptophans or tyrosines.

conjugated by qBBr (*Figure 2A, B*). Therefore in the case of R2 we cannot assess precisely how well the qBBr replaces the arginine, but it is important to note that the fluorescence, as opposed to gating currents, will reflect the position of the qBBr. The above results suggest that qBBr seems to act as a faithful mimic of the movement of the gating charge of R1, and may also mimic the movement of R2, albeit with energetic differences.

Upon depolarization, the fluorescence signal of R1C-qBBr decreased dramatically (*Figure 2C*, top), while the corresponding fluorescence signal of R2C-qBBr did not (*Figure 2C*, bottom). A reduction in

qBBr fluorescence is likely due to the presence of a tryptophan, or possibly a tyrosine. Since the fluorescence signal is lower when the R1C-qBBr is in the active state than when it is in the resting state, the quenching residue should be near the active state of R1. Based on a homology model of a Kv channel crystal structure in the active state (*Chen et al., 2010*; *Long et al., 2005*), R1 should reside near an endogenous tryptophan (W454) at the extracellular side of S6 (*Figure 2D*). Additionally, it has been reported that in the active state R1 comes into close proximity to this region of the channel (*Elinder et al., 2001*; *Lainé et al., 2003*). The mutation of this tryptophan to a nonquenching alanine abolished the fluorescence reduction that we observed during activation (*Figure 2E*, top). The mutation of this tryptophan to a phenylalanine also abolished the native tryptophan-driven voltage-dependent fluorescence (*Figure 2E*, bottom). However, as this mutation reduced expression, we proceeded with a W454A background. Therefore, our optical data support the idea that upon activation, R1C-qBBr comes near the pore domain where its fluorescence is quenched by W454, while R2C-qBBr does not.

These results are in agreement with crystallographic and functional data (*Chen et al., 2010*; *Elinder et al., 2001*; *Lainé et al., 2003*; *Long et al., 2005*) and suggest that R2C-qBBr and R1C-qBBr may mimic the motion of the native gating charges.

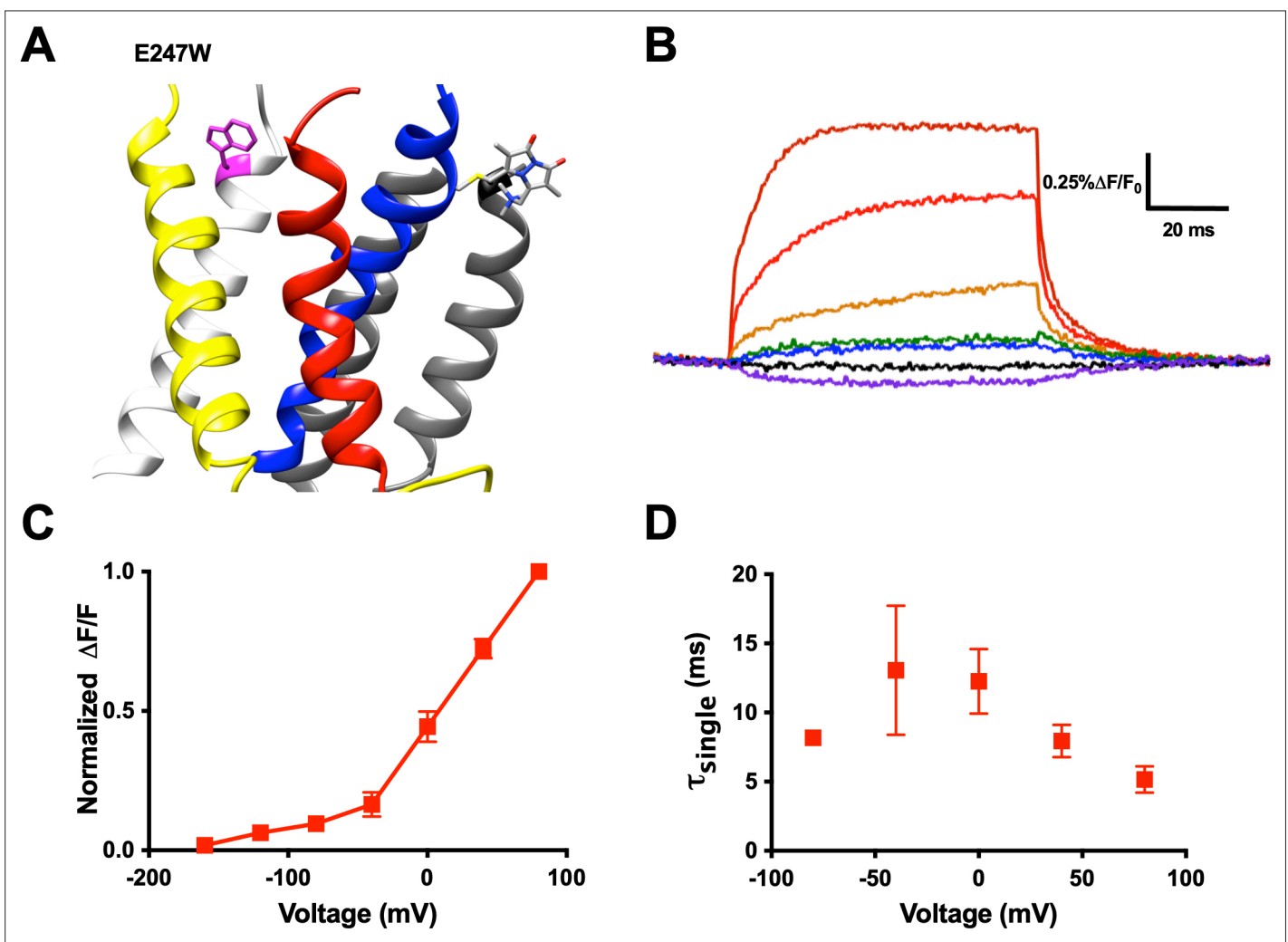

**Figure 3.** An exogenously substituted tryptophan (E247W) produces voltage-dependent fluorescence changes in R1C-qBBr:W454A. (**A**) Homology structure of Shaker construct, R1C-qBBr:W454A;E247W demonstrating the tryptophan placement within the protein. (**B**) A representative activation family of fluorescence traces. (**C**) A normalized $\Delta F/F_0$ fluorescence versus voltage curve. Normalization was performed for each oocyte separately (*n* = 4). (**D**) A single exponential $\tau$ of fluorescence upon activation (*n* = 3). Data are shown as mean ± standard error of the mean (SEM).

The online version of this article includes the following source data for figure 3:

**Source data 1.** Source data for normalized $\Delta F/F_0$ versus voltage in *Figure 3C* and single exponential $\tau_{act}$ of fluorescence in *Figure 3D*.

## An exogenously substituted tryptophan (E247W) produces voltage-dependent fluorescence changes in R1C-qBBr:W454A

To map the quenching interactions of R1 and R2 using qBBr, we created a series of constructs based on the R1C:W454A and the R2C backgrounds. We used these two constructs as backgrounds because they only showed a very small depolarization-induced increase in fluorescence that had a fast time course with no voltage-dependent kinetics (*Figure 2E*); we were unable to link their fluorescence to any endogenous tyrosine or tryptophan (*Figure 2—figure supplement 1*). As this residual fluorescence change was not observed in the presence of tryptophan-induced quenching in the R1C-qBBr:W454 construct, we hypothesized that substituting a tryptophan into other regions of the channel likely to come near a gating charge would similarly alter the qBBr fluorescence signal in a way that would dominate over the residual fluorescence signal.

As an example, we substituted a tryptophan at position 247 of the R1C-W454A construct, generating R1C-qBBr:W454A;E247W (*Figure 3A*). In response to changes in membrane potential, the exogenous, substituted tryptophan produced voltage-induced fluorescence changes (*Figure 3B*) that were markedly different from the background fluorescence changes of R1C-qBBr:W454A (*Figure 2E*, top). If R1C-qBBr is closer to the tryptophan at position 247 in the resting state than in the active state, we should observe an increase in fluorescence upon activation (*Figure 1C*, top). The experiment shows that during activation, E247W produces a clear unquenching, or increase in fluorescence, of R1C-qBBr (*Figure 3B*). Therefore, R1C-qBBr moves away from position 247 upon activation.

Examining the fluorescence signal more closely, we see that it has many properties reflecting movement of the voltage sensor. Just as the gating currents produced by the voltage sensor are voltage sensitive, the fluorescence signal of R1C-qBBr:W454A;E247W has a voltage-sensitive response. From our family of fluorescence traces (*Figure 3B*), we can generate a change in fluorescence versus voltage (FV) curve and observe the amount of fluorescence quenching or unquenching due to changes in voltage (*Figure 3C*). Moreover, the voltage sensitivity of the fluorescence is also seen in the kinetics of the fluorescence traces (*Figure 3D*).

## Activation pathways mapped by qBBr and substituted tryptophan-induced fluorescence quenching

We generated numerous constructs with an exogenous, substituted tryptophan that produced voltage-induced fluorescence changes in R1C-qBBr:W454A or R2C-qBBr that were distinct from the background fluorescence changes and from each other (*Figure 3B, C* and *Figure 4A, B*). A decrease in fluorescence upon depolarization suggests that the R1C- or R2C-qBBr comes closer to the inserted tryptophan residue during the transition from the resting to the active state, while an increase in fluorescence upon depolarization suggests that the gating charge moves away from that residue during activation (*Figure 1C*). Therefore, by pooling the data from numerous constructs, each with a single tryptophan, we can map the activation pathways of R1 and R2 (*Figures 4A and 5A*).

In response to a depolarizing pulse, R1C-qBBr fluorescence was quenched by the endogenous tryptophan at residue 454, as well as by a tryptophan substituted for the tyrosine at the extracellular side of the S5 at position 415. R1C-qBBr moved away from tryptophans inserted at F244W, E247W, I320W, and T326W (*Figure 4A, B*). Interestingly, we found no appreciable changes in the fluorescence signal of F290W or L294W in response to depolarizing pulses (*Figure 4A, B*). The small fast fluorescence signal seen for these two constructs is reminiscent of that of R1C-qBBr:W454A (*Figure 2E* and *Figure 2—figure supplement 1*), that is, the residual fluorescence signal that is small and has fast voltage-independent kinetics. This residual fluorescence signal is not observed in qBBr-labeled oocytes that were not injected with mRNA. Together these findings are consistent with a pathway for R1 that consists of both an intracellular to extracellular translation, tilted by about 30° with respect to the normal of the membrane plane, and a rotation (*Figure 4B*; *Campos et al., 2007*; *Delemotte et al., 2011*; *Glauner et al., 1999*; *Tombola et al., 2007*; *Wisedchaisri et al., 2019*; *Yarov-Yarovoy et al., 2006*). Specifically, R1C-qBBr seems to reside near the pore domain in the active state, as observed in the crystal structure (*Chen et al., 2010*). It reaches this position from a resting position closer to F244, I320, T326, and E247, each of which reside within the VSD and produce a fluorescence unquenching of R1C-qBBr upon activation.

Substitution of W residues along the pathway of R2 reveals that upon activation R2C-qBBr moves closer to Y415W, F416W, and T326W, while moving away from I241W, F244W, I287W, and L294W

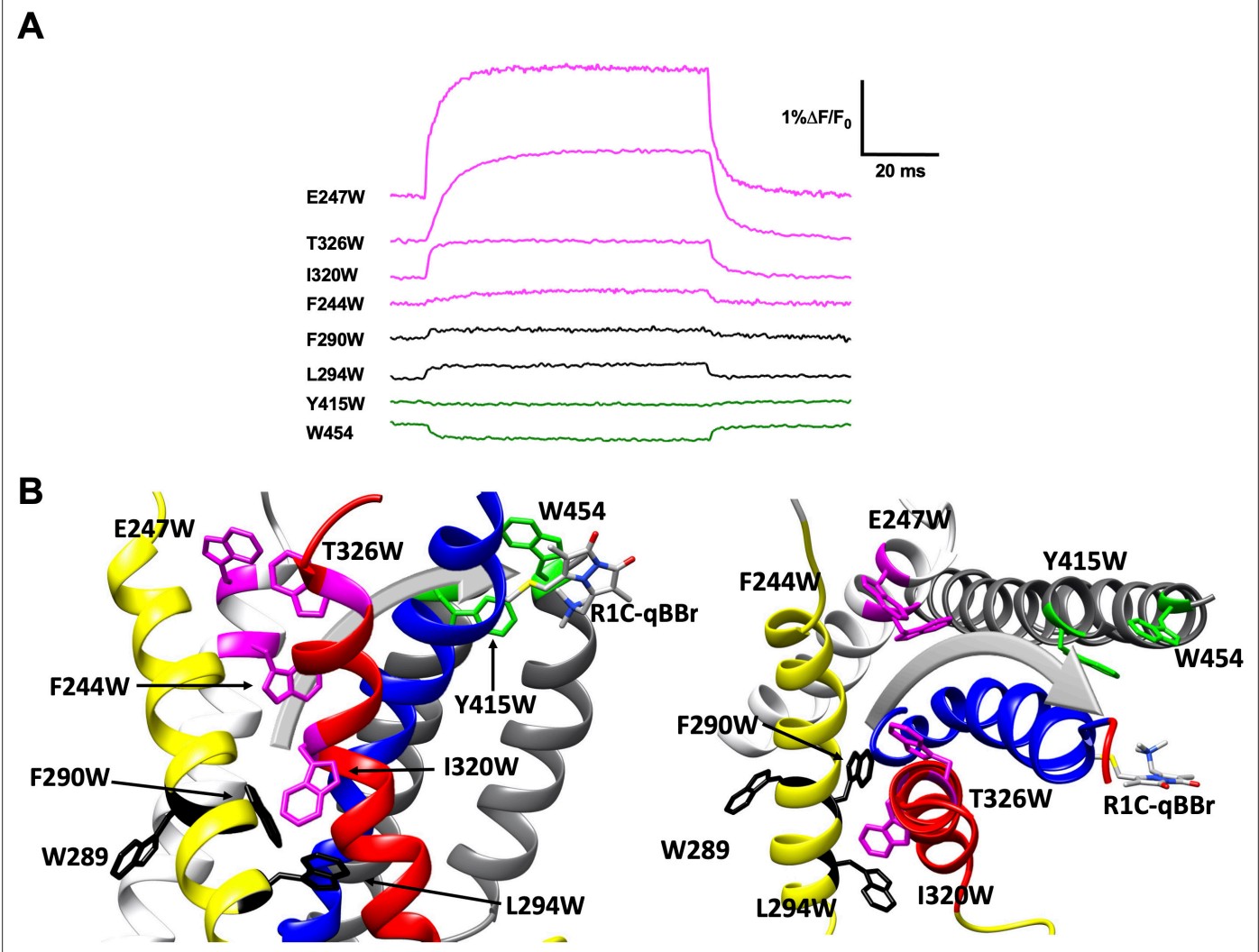

**Figure 4.** Activation pathway of R1C-qBBr:W454A mapped by several individually substituted tryptophans. (**A**) Representative qBBr traces at +80 mV. As qBBr moves closer to a W, the W quenches (green) the qBBr fluorescence (Y415W and W454). When it moves further away from a W, the qBBr fluorescence is unquenched (pink; F244W, E247W, I320W, and T326W). Some W mutations have no effect (black; 289W, F290W, and L294W). (**B**) A summary structure of activation of R1C-qBBr, with side (left) and extracellular (right) views of the voltage-sensing domain (VSD). The activation pathway based on the W quenching/unquenching for R1C-qBBr is marked by a gray arrow.

(*Figure 5A, B*). The fluorescence of R2C-qBBr:L294W is kinetically distinct from that of R2C-qBBr, so it is likely that this is a true unquenching we observe, as opposed to R1C-qBBr:W454A;F290W, which appears identical to R1C-qBBr:W454A (*Figure 5—figure supplement 1*). Examining the quenching of R2C-qBBr from T326W, Y415W, and F416W, we see that in the active state, R2C-qBBr resides between the extracellular side of the S3 and S5 segments. As opposed to R1C-qBBr, R2C-qBBr does not encounter the S6 segment upon activation. In the resting state, the fluorescence results suggest that R2C-qBBr is stable near F290, unquenching from tryptophans inserted at L294, I287, F244, and I241. As seen with R1, there is a rotation in the movement of R2; however, the rotation of R2 is less evident.

Interestingly, the fluorescence responses differ qualitatively between the two gating charges for tryptophans substituted at L294, T326, and W454. Thus, qBBr mapping reveals that R1 and R2 travel distinct paths in relationship to the VSD during activation. Notable exceptions to this include Y415, W289, and F244, which produce similar responses from both R1C and R2C-qBBr. As it faces the lipid membrane, it is unsurprising that W289 does not quench either R1C- or R2C-qBBr; however, this suggests that the S2 in which it resides is unlikely to undergo any large rotations that would expose

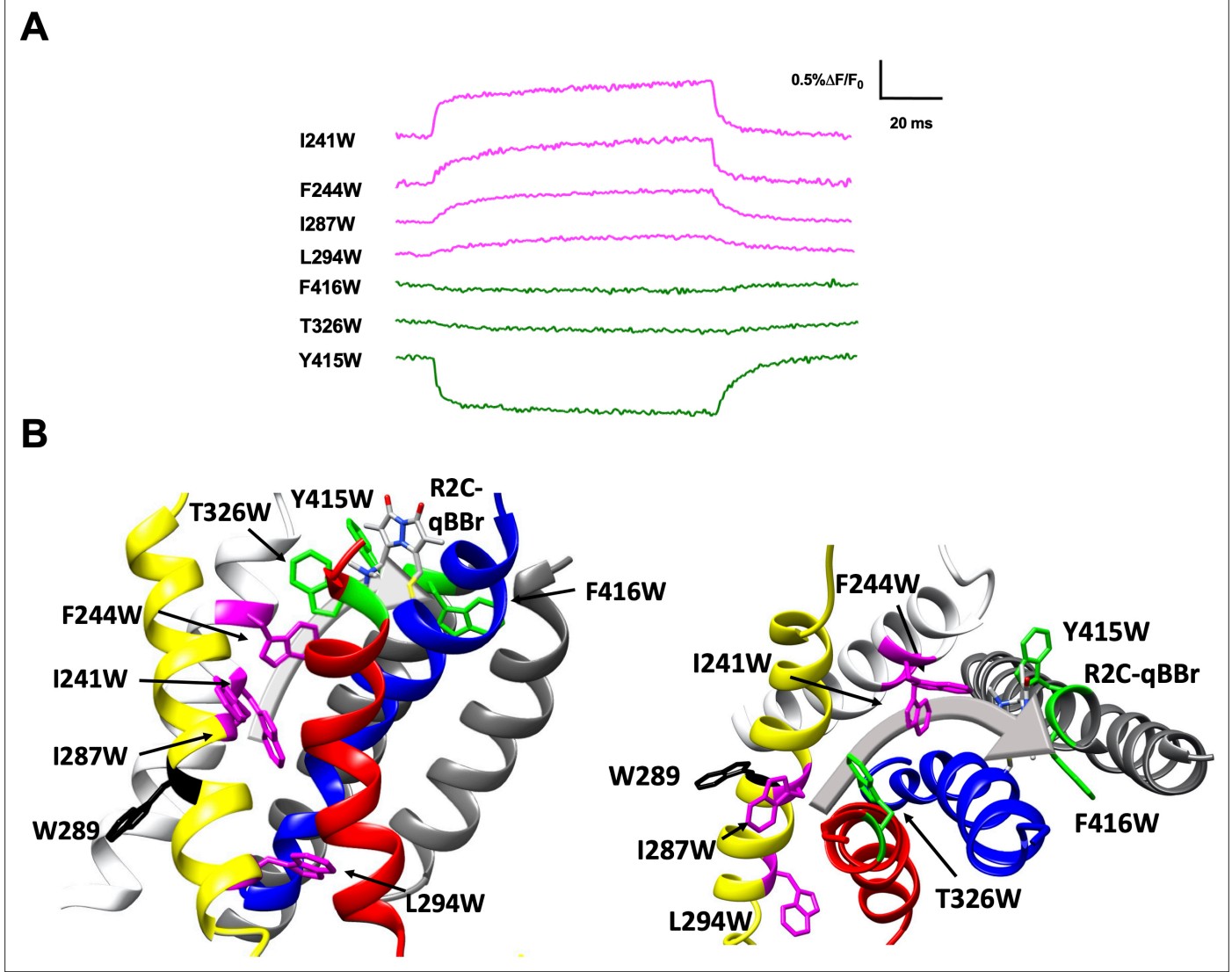

**Figure 5.** Activation pathway of R2C-qBBr mapped by several individually substituted tryptophans. (**A**) Representative qBBr traces at +80 mV. As qBBr moves closer to a W, the W quenches (green) the qBBr fluorescence (T326W, Y415W, and F416W). When it moves further away from a W, the qBBr fluorescence is unquenched (pink; I241W, F244W, I287W, and L294W). Residue W289 (black) does not affect the fluorescence. (**B**) A summary structure of activation of R2C-qBBr with side (left) and extracellular (right) views of the voltage-sensing domain (VSD). The activation pathway based on the W quenching/unquenching for R2C-qBBr is marked by a gray arrow.

The online version of this article includes the following figure supplement(s) for figure 5:

**Figure supplement 1.** Effect of L294W in both R1C-qBBr:W454A and R2C-qBBr.

this residue to the gating charges. Y415W quenches both R1 and R2, underscoring the activated position of both these residues near the extracellular surface of the membrane. The neighboring residue F416W quenched R2C-qBBr and did not express with R1C-qBBr. F416 has previously been shown to bridge to R1 (*Conti et al., 2016*; *Lainé et al., 2003*; *Phillips and Swartz, 2010*) and R2 (*Conti et al., 2016*). Finally, F244 has been shown to interact with R2 in an intermediate state (*Lacroix et al., 2012*), and has been proposed to interact with R3 as a critical stabilizer of the active state of the voltage sensor (*Lacroix et al., 2014*). Our fluorescence data from both R1C-qBBr and R2C-qBBr are consistent with a model in which upon activation both R1 and R2 undergo a tilted translation from the intracellular to the extracellular side of the membrane, together with a rotation (*Figures 4B and 5B*, *Table 1*; *Catterall, 1986*; *Guy and Seetharamulu, 1986*). The discrete and unique pathways mapped out by these charges provide new levels of clarity into how these charges move distinctly from each other.

**Table 1.** Summary $\Delta F/F_0$ of qBBr constructs.

| Construct | $\Delta F/F_0$ ± SEM at 80 **mV** | Fluorescence effect | *N* |
|---|---|---|---|
| Sh WF | N/A | N/A | 3 |
| Sh WF R1C-qBBr | −0.154 ± 0.0172 | Quench | 4 |
| Sh WF R1C-qBBr:W454A | 0.206 ± 0.037 | No effect | 5 |
| Sh WF R1C-qBBr:W454A;F244W | 0.191 ± 0.026 | Unquench | 3 |
| Sh WF R1C-qBBr:W454A;E247W | 0.930 ± 0.205 | Unquench | 8 |
| Sh WF R1C-qBBr:W454A;F290W | 0.0946 ± 0.0323 | No effect | 4 |
| Sh WF R1C-qBBr:W454A;L294W | 0.127 ± 0.086 | No effect | 4 |
| Sh WF R1C-qBBr:W454A;I320W | 0.247 ± 0.095 | Unquench | 4 |
| Sh WF R1C-qBBr:W454A;T326W | 0.448 ± 0.167 | Unquench | 6 |
| Sh WF R1C = qBBr:W454A;Y415W | −0.147 ± 0.015 | Quench | 8 |
| | | | |
| Sh WF R2C-qBBr | 0.168 ± 0.0241 | No effect | 5 |
| Sh WF R2C-qBBr:I241W | 0.261 ± 0.132 | Unquench | 4 |
| Sh WF R2C-qBBr:F244W | 0.454 ± 0.171 | Unquench | 4 |
| Sh WF R2C-qBBr:I287W | 0.493 ± 0.156 | Unquench | 4 |
| Sh WF R2C-qBBr:L294W | 0.183 ± 0.036 | Unquench | 3 |
| Sh WF R2C-qBBr:T326W | −0.094 ± 0.017 | Quench | 3 |
| Sh WF R2C-qBBr:Y415W | −0.851 ± 0.123 | Quench | 9 |
| Sh WF R2C-qBBr:F416W | −0.0448 ± 0.098 | Quench | 4 |

Summary $\Delta F/F_0$ **of qBBr constructs** – The $\Delta F/F_0$ for all constructs and the effect of the construct's tryptophan on qBBr fluorescence at a depolarizing pulse to +80 mV.
SEM = standard error of the mean..

## VSD deactivation transitions are followed by qBBr

Previous studies have discussed the asymmetry in activation and deactivation currents, with deactivation currents being slower than those of activation (*Labro et al., 2012*; *Lacroix and Bezanilla, 2012*; *Lacroix et al., 2011b*; *Zagotta et al., 1994b*). Specifically, deactivation currents are slowed down in a voltage-dependent manner which corresponds with pore opening (*Labro et al., 2012*; *Lacroix et al., 2011b*; *McCormack et al., 1994*). Therefore, gating currents frequently display different voltage dependence and kinetics during activation (*Figure 6A*, left) and deactivation (*Figure 6A*, right). If qBBr is behaving as a fluorescent gating charge, we would expect its fluorescence to display voltage dependence shifts similar to those seen in the underlying gating currents. The comparison of the activation (QVs) to the deactivation QVs shows that during deactivation there are leftward shifts in the R1C-qBBr:W454A constructs E247W and T326W, but not in I320W (*Figure 6B*), and in the R2C-qBBr constructs F244W and I287W (*Figure 6—figure supplement 1*). In each of these constructs, the FV also shows a leftward shift in deactivation compared to activation. Therefore, these data show that qBBr appears capable of broadly recapitulating deactivation. We see a similar phenomenon in comparing fluorescence kinetics during activation to those during deactivation; in most constructs examined, the deactivation kinetics were slower than the activation kinetics, as expected. The one exception was R1C-qBBr:W454A;T326W. The T326W mutation slows activation much more strongly than it slows deactivation (*Hong and Miller, 2000*). Thus, overall, we have strong evidence that the qBBr fluorescence follows deactivation transitions as well as activation transitions, and that qBBr follows the different energetic paths experienced by the voltage sensor during deactivation and activation.

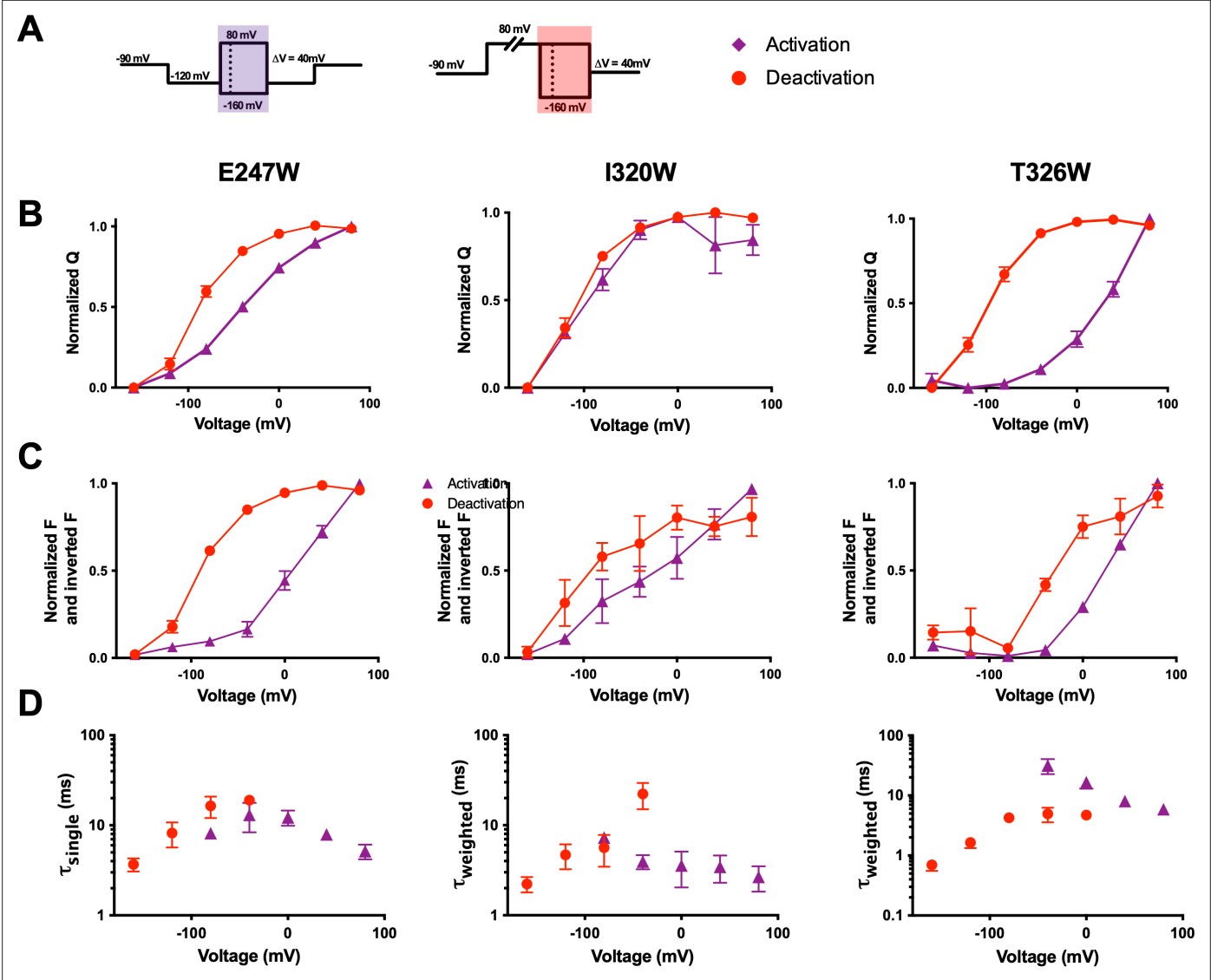

**Figure 6.** Voltage-sensing domain (VSD) deactivation transitions are followed by qBBr. (**A**) Activation (purple) and deactivation (red) protocols. Shading indicates where gating and fluorescence were measured for (**B–D**). (**B**) Comparison of activation (purple triangles) and deactivation (red circles) QVs for R1C-qBBr:W454A;E247W (left, activation $n$ = 4, deactivation $n$ = 9), R1C-qBBr:W454A;I320W (center, activation $n$ = 4, deactivation $n$ = 5), and R1C-qBBr:W454A;T326W (right, activation $n$ = 5, deactivation $n$ = 5). (**C**) As in (**B**), but a comparison of activation and deactivation FVs, rather than QVs. (**D**) Comparison of single exponential $\tau_{act}$ (red circles, $n$ = 3) to $\tau_{dea}$ (purple triangles, $n$ = 4) of R1C-qBBr:W454A;E247W (left). Weighted $\tau_{act}$ (red circles, $n$ = 3) to $\tau_{dea}$ (purple triangles, $n$ = 5) of R1C-qBBr:W454A;I320W (center). Weighted $\tau_{act}$ (red circles, $n$ = 3) to $\tau_{dea}$ (purple triangles, $n$ = 3) of R1C-qBBr:W454A;T326W (right). Data are shown as mean ± standard error of the mean (SEM).

The online version of this article includes the following source data and figure supplement(s) for figure 6:

**Source data 1.** Source data for normalized QVs in *Figure 6A*, normalized FVs in *Figure 6B*, and single or weighted $\tau$ of fluorescence in *Figure 6C*.

**Figure supplement 1.** The voltage-sensing domain (VSD) deactivation path of R2C-qBBr differs from that of activation.

**Figure supplement 1—source data 1.** Source data for normalized QVs in *Figure 6—figure supplement 1A*, normalized FVs in *Figure 6—figure supplement 1B*, and single or weighted $\tau$ of fluorescence in *Figure 6—figure supplement 1C*.

**Figure supplement 2.** R1C-qBBr fluorescence visualizes the relaxed state.

**Figure supplement 2—source data 1.** Source data for normalized QVs in *Figure 6—figure supplement 2B* and normalized FVs in *Figure 6—figure supplement 2C*.

One observation of potential interest is that for the R1C-qBBr constructs, the FV does not always perfectly align with the corresponding QV. For example, in the R1C-qBBr:W454A;E247W construct, the FV of deactivation is left shifted in comparison to that of activation, in good agreement with the QVs. However, the activation FV is right shifted in comparison to the activation QV. Similarly, the deactivation FV of R1C-qBBr:W454A;T326W is right shifted in comparison to the deactivation QV. These differences likely stem from the QV measuring every gating charge in relationship to the electric field, while the FV measures the relationship between a single gating charge and a particular residue in the VSD. Thus, the R1C-qBBr seems to require more energy to move past, or away from, E247W during activation, compared to the bulk movement of the voltage sensor, and the charge seems to require less energy to move toward T326W during deactivation.

Additionally, we measured the fluorescence of R1C-qBBr:W454A;E247W and R1C-qBBr:W454A;T326W during a return to hyperpolarized potential after a prolonged depolarizing pulse (800 ms, +40 mV). This produced a left-shifted FV compared to FVs produced by deactivation and activation of the same construct (*Figure 6*, *Figure 6—figure supplement 2*). This further left-shifted FV may be related to the relaxed state, or mode shift, reached by many S4-based voltage sensors (*Bezanilla et al., 1982a*; *Bruening-Wright and Larsson, 2007*; *Villalba-Galea et al., 2008*). Although additional investigation will be required, the relaxed state is described by a leftward shift in QVs and a slowing of kinetics upon a return to hyperpolarized potential after a prolonged depolarization (*Lacroix et al., 2011b*; *Villalba-Galea et al., 2008*).

## R1C–qBBr interaction with F290 provides the basis of the Cole–Moore shift

In addition to uncovering information about transitions to and from the resting and active states, qBBr tracking provides novel insight into the Cole–Moore shift. The Cole–Moore shift describes a phenomenon where following a prolonged hyperpolarizing prepulse, a depolarizing pulse elicits a conductance that has a longer time lag, or delay before beginning, than when there is no prepulse; this lag prolongs as the prepulse is made longer and more negative (*Cole and Moore, 1960*; *Hoshi and Armstrong, 2015*; *Stefani et al., 1994*; *Taylor and Bezanilla, 1983*). The Cole–Moore shift is also seen in the gating currents underlying the conductance (*Bezanilla et al., 1982b*; *Bezanilla et al., 1994*). The Cole–Moore shift can be elicited in several ways. It is typically measured following a prepulse of increasing hyperpolarized voltage before a depolarization (*Figure 7B*, top left) or increasingly long hyperpolarizing prepulses (*Figure 7B*, bottom left). The classical explanation for the Cole–Moore shift is that a hyperpolarization populates closed states that are further removed from the open state, thus delaying the opening as the channel has to traverse more states before opening (*Cole and Moore, 1960*). However, the molecular basis of the Cole–Moore shift remains unknown (*Hoshi and Armstrong, 2015*).

While measuring R1C-qBBr:W454A;F290W fluorescence, we observed no appreciable fluorescence signal above the residual fluorescence in response to depolarizing pulses from −120 mV (*Figure 7A*). However, in response to a hyperpolarizing pulse to −160 from −120 mV, we observed a small, markedly slow reduction in fluorescence (*Figure 7A*). This fluorescence change did not have the same kinetics of a small measurable gating current for the same pulse protocol (*Figure 7—figure supplement 1*). When a protocol used to induce and measure a Cole–Moore shift (*Figure 7B*, left bottom) was applied, the R1C-qBBr:W454A;F290W fluorescence response was larger than the residual response to depolarizing pulses seen in *Figure 7A*, and it became slower as the duration of the hyperpolarizing pulse increased (*Figure 7C*), as expected if this fluorescence is associated with the Cole–Moore shift of the channel.

Previous studies have shown that at negative potentials R1C in the closed state can spontaneously link to I287C, which is a full turn above F290 (*Campos et al., 2007*). Here, we observe that at extreme hyperpolarizing potentials (−160 mV) R1C-qBBr moves near F290W, producing a slow quenching of fluorescence. We explain this as a result of moving the VSD with a strong hyperpolarization to such an extreme intracellular position that we populate other closed states of the voltage sensor that are responsible for the Cole–Moore shift. These closed states are tracked by the fluorescence change produced by an interaction between F290W and R1C-qBBr. A similar rarely observed closed state predicted by metal–ion bridges, in which R1 transitions from a position extracellular to F290 to a position intracellular to F290, was also suggested to provide a potential explanation of the Cole–Moore

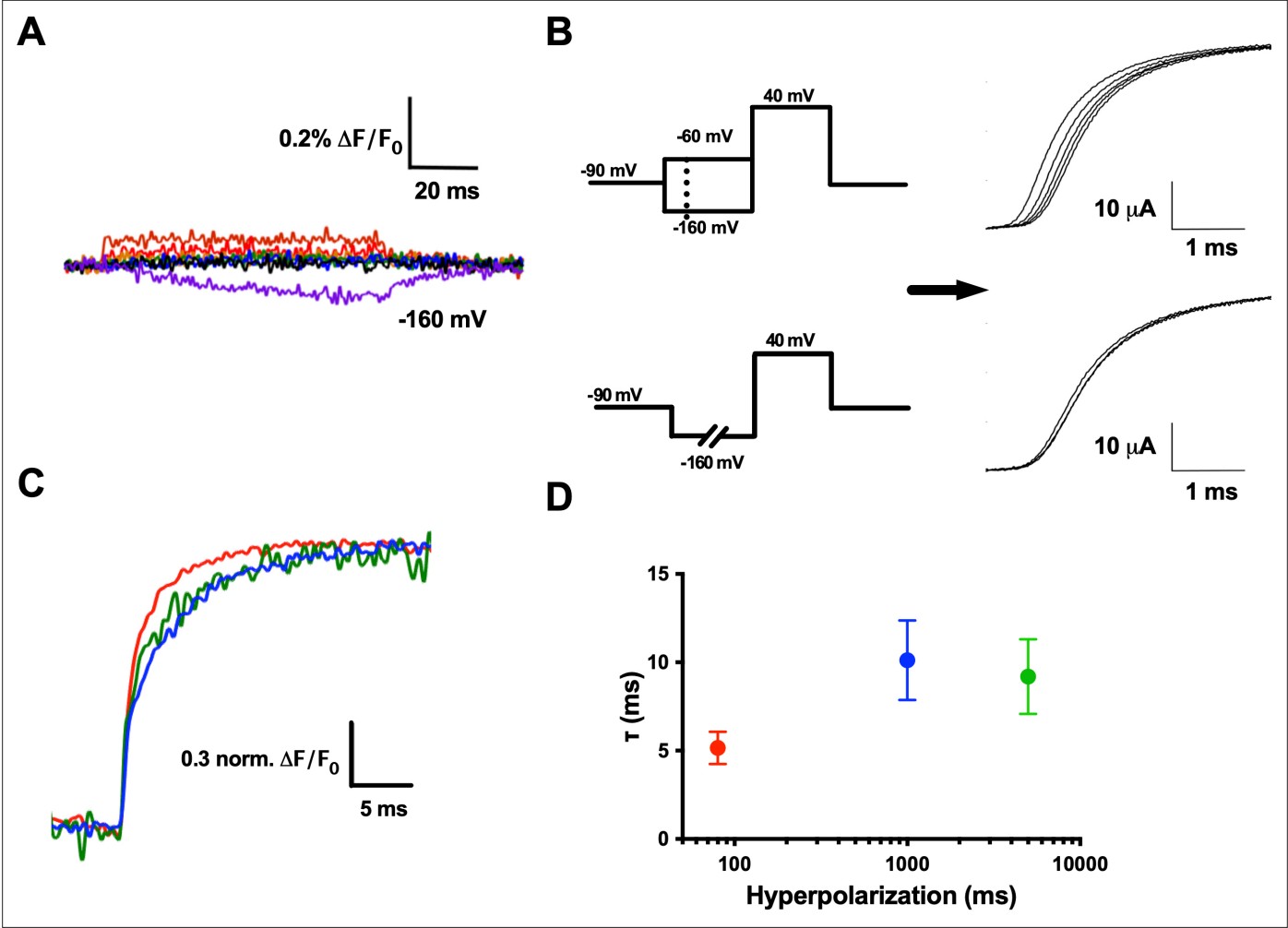

**Figure 7.** R1C–qBBr interaction with F290 provides the basis of the Cole–Moore shift. (**A**) A family of fluorescence traces for R1C-qBBr:W454A;F290W from a pulse protocol as in (6A). Note the slow fluorescence response when hyperpolarized to −160 mV (purple) and the residual signals during depolarizations. (**B**) Two pulse protocols that induce a Cole–Moore shift: a variable voltage pulse (−80 to −160 mV, every −20 mV) before a depolarization (top) or a prolonged hyperpolarization pulse (5, 1000, and 3000 ms) before depolarization (bottom) and the resulting Cole–Moore shifts of representative Shaker ionic currents (right). (**C**) Representative normalized fluorescence traces for R1C-qBBr:W454A;F290W using the pulse protocol from (B, bottom),with a −160 mV prepulse with a variable duration for 80 ms (red), 1 s (blue), and 5 s (green). (**D**) Comparison of the $\tau_{weighted}$ of the change in qBBr fluorescence in (C) (80 ms pulses, $n$ = 4; 1 s pulses, $n$ = 5; 5 s pulses, $n$ = 4). Data are shown as mean ± standard error of the mean (SEM).

The online version of this article includes the following source data and figure supplement(s) for figure 7:

**Source data 1.** Source data for weighted fluorescence $\tau$ in **Figure 7D**.

**Figure supplement 1.** R1C-qBBr:W454A;F290W hyperpolarization-induced gating charge and fluorescence signal time course.

shift (**Henrion et al., 2012**). Finally, our finding that R1 resides extracellular to the hydrophobic plug of the VSD may help explain why R1 is reasonably tolerable to substitution by qBBr and by the neutral analog citrulline (**Infield et al., 2018**).

## Voltage-sensitive membrane protein CiVSP is interrogable by qBBr

Our technique presented here allows for the interrogation of other VSD movements. Another excellent candidate for qBBr mapping is CiVSP (**Murata et al., 2005**). With the reported crystal structure (**Li et al., 2014**) we were able to begin qBBr mapping of this VSD (**Figure 8A**). Using the CiVSP R217Q background (**Dimitrov et al., 2007**) we mutated the first gating charge, R223, to a cysteine and labeled it with qBBr. The CiVSP R1C-qBBr showed voltage-dependent fluorescence changes (**Figure 8B**, top). As with Shaker R1C-qBBr, we were able to identify the source of the fluorescent change. By mutating tyrosine Y206 to an alanine, we abolished the voltage-dependent fluorescence (**Figure 8B**, bottom),

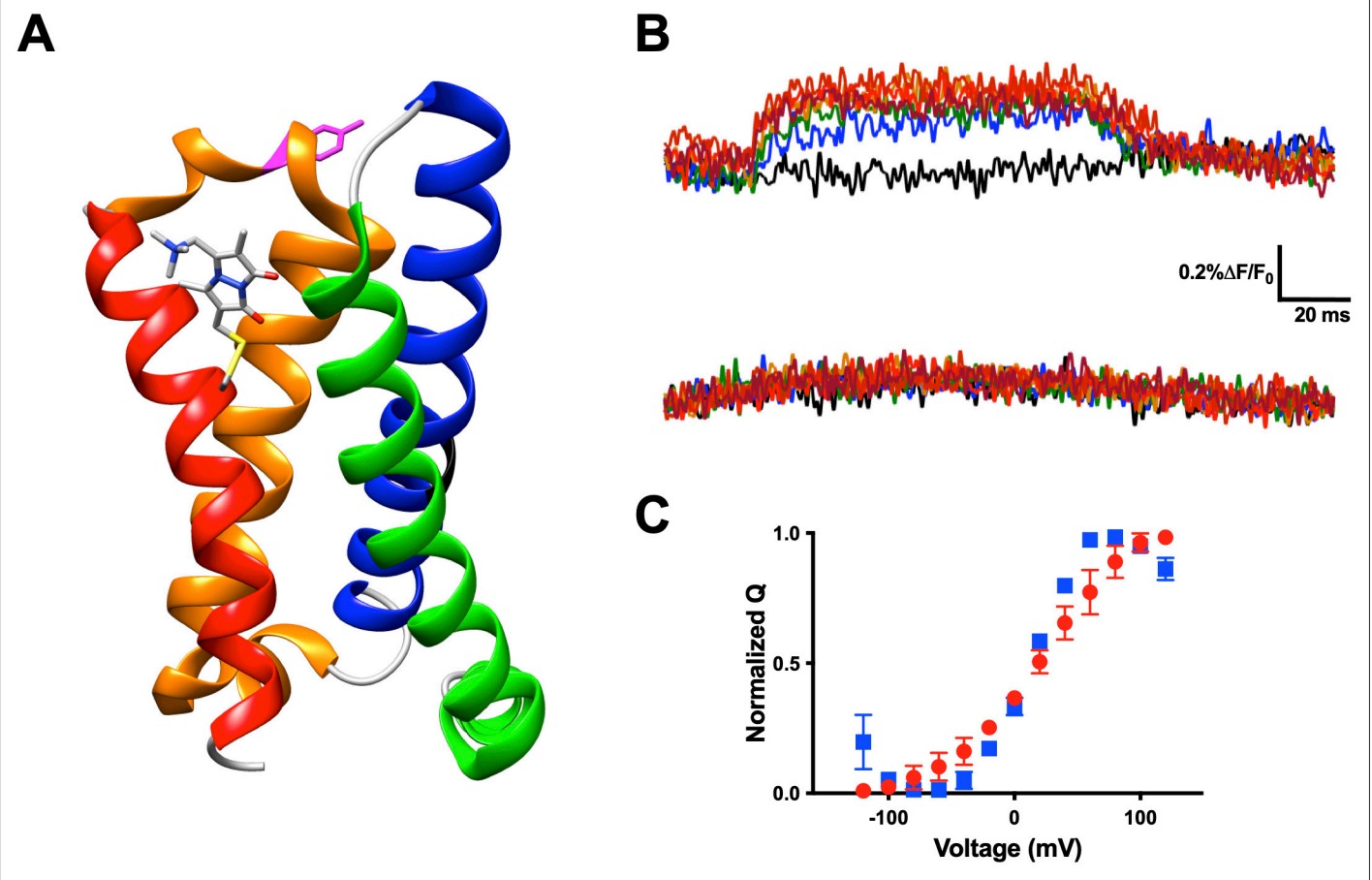

**Figure 8.** Voltage-sensitive membrane protein CiVSP is interrogable by qBBr. (**A**) CiVSP structure (PDB: 4G7V) with qBBr attached at R1C and highlighting residue Y206. Transmembrane domains (S1–S4) are colored with S1, green; S2, blue; S3, orange; S4, red. (**B**) Representative fluorescence traces of CiVSP R217Q R1C-qBBr (top) and CiVSP R217Q R1C-qBBr:Y206A (bottom). (**C**) Normalized *QV* curves comparing CiVSP R217Q R1C-qBBr (red circles, *n* = 4) and CiVSP R217Q R1C-qBBr:Y206A (blue squares, *n* = 3). Data are shown as mean ± standard error of the mean (SEM).

The online version of this article includes the following source data for figure 8:

**Source data 1.** Source data for normalized QV in *Figure 8C*.

while the voltage dependence of CiVSP with the Y206A mutation remained intact (*Figure 8C*). This result is a starting point for investigating the movement of the gating charges of this VSD.

## Discussion

In this paper, we have demonstrated a technique that allows tracking of a gating charge surrogate in the pathway of a voltage sensor using qBBr. We have obtained new information on the steady state positions of the two most extracellular gating charges in Shaker (*Figures 4 and 5*). We propose that this is a flexible tool that should prove readily applicable to other voltage-sensitive proteins. As we discuss the details of the movements we observed, we will point out some of the strengths and limitations of the technique.

### Spatial resolution

In principle, one could expect qBBr mapping to be extremely precise in its spatial resolution. The photoinduced electron transfer (PET) mechanism that produces the quenching of bimane by tryptophan requires van der Waals contact between the bimane and the tryptophan; as a result, quenching interactions require center-to-center distances below 10 Å (*Doose et al., 2009*; *Lakowicz, 2006*). PET is distinct from resonance energy transfer based quenching; the latter is highly dependent on the mutual orientation of the electric dipole moments of the donor and acceptor, while the former

is influenced by the orientation of the electron orbitals (*Doose et al., 2009*; *Lakowicz, 2006*). At a spatial scale spanning multiple angstroms composed of multiple electron clouds, bulk anisotropy is unlikely.

Converting this high degree of spatial resolution into a biological system is challenging, however. Tryptophan-induced quenching (TrIQ) and tyrosine-induced quenching (TyrIQ) of bimane dyes have been proposed as methods for measuring molecular distances (*Brunette and Farrens, 2014*; *Mansoor et al., 2002*; *Mansoor et al., 2010*). The maximum alpha carbon to alpha carbon distance over which TrIQ occurs with qBBr has been found to be around 10–15 Å (*Brunette and Farrens, 2014*; *Mansoor et al., 2002*; *Mansoor et al., 2010*). Further complicating matters is that alpha carbon to alpha carbon distances do not perfectly capture the quenching efficiency, as different rotamers of the bimane dye and its quenching tryptophan will change the distance between side chains and, consequently, the quenching efficiency. As an example, in our system, the R2C-qBBr side chain is quenched by tryptophans substituted at two adjacent residues: Y415 and F416. Despite the alpha carbon distances differing by <4 Å, the magnitude of quenching from Y415W is much greater than that from F416W ($\Delta F/F_0$ = −0.85% ± 0.12% versus −0.045% ± 0.10%). This suggests that the favored rotamers of Y415W could bring the quenching tryptophan into much closer contact with R2C-qBBr than the favored rotamers of F416W do. The dependence of qBBr quenching on the rotameric orientation of the dye and the quencher underscores the importance of recording fluorescence changes using multiple constructs with tryptophan substituted at distinct sites.

Absolute calibration of the magnitude of fluorescence quenching to a particular distance is also thwarted by variability in background fluorescence. Developments in chemically reducing oocyte fluorescence (*Lee and Bezanilla, 2019*) may help surmount this obstacle in the future. Currently, we can use the technique presented here to investigate the insertion of a quencher in the putative path: when a fluorescence change occurs during a voltage pulse, we can infer the charge is moving with respect to the quencher while if the fluorescence signal does not change, it means that either the quencher is far from the charge's path or there is no movement with respect to the quencher.

Take as an example qBBr moving from a position adjacent to W to a distance of 7 Å away from the quencher during a depolarization. The qBBr–W distance is always shorter than the 10 Å known to be the qBBr–W quenching distance. However, the degree of quenching will vary from extremely high when qBBr is close to W to less as it moves to its final position. The change in fluorescence may be only 1–2%, but this change still reveals that the qBBr-quencher distance is increasing. In other words, we infer that it moved away even though we do not know the exact distance.

In both our R1C-qBBr and R2C-qBBr datasets, the bimane fluorescence is seen to unquench from multiple tryptophans substituted into the VSD at residues that span a carbon alpha to carbon alpha distance of roughly 18–23.5 or 13–18.5 Å in the axis perpendicular to the membrane. Two important points prevent us from concluding that the S4 backbone is moving this distance. First, the length of the tryptophan and qBBr side chains is about 5–6 Å and they are capable of entering into a PET quenching interaction with each other at a distance of at least a few angstroms. Second, our data do not show that in the resting state, qBBr is in close contact with each residue it unquenches from upon activation. Indeed, the movement of qBBr past a tryptophan will produce an unquenching effect as long as its final distance from that tryptophan is greater than its initial distance from that tryptophan (*Figure 1C*, *Figure 1—figure supplement 1*). Consequently, spatial resolution and constraints on, for example, the resting state of a particular individual gating charge, are improved by increasing the number of substituted quenching residues found to interact or not interact with the gating charge. Thus, even though the interpretation is qualitative rather than quantitative, our method can constrain the resting and active positions of a particular gating charge and generate its putative transition pathway by collecting data from many interacting and noninteracting residues.

## Kinetics of trajectories are different for each charge and depend on initial conditions

As laid out in *Equation 2*, in dynamic qBBr–W quenching, a two-state model predicts that the kinetics of the fluorescence is independent of the position of qBBr with respect to the quencher. If qBBr moves between several discrete positions, then kinetics of the fluorescence is affected by the relative distance with respect to the quencher (*Equation 3*). In a three-state model, the general prediction is a time course of fluorescence with increases and decreases during the voltage pulse that moves the

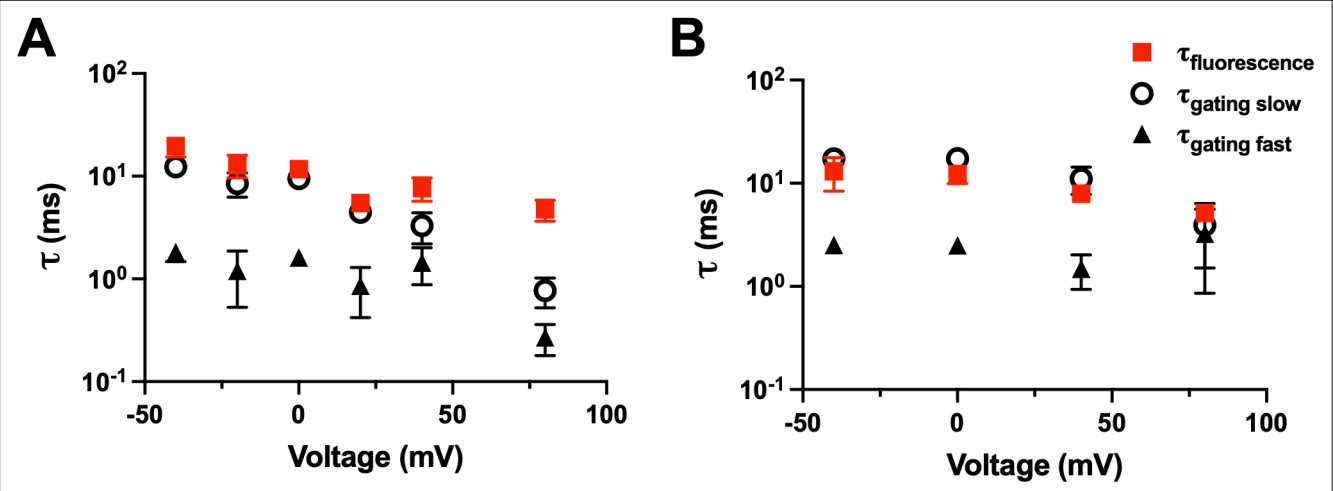

**Figure 9.** Fluorescence kinetics of qBBr. (**A**) Comparison of the single exponential tau of fluorescence of R1C-qBBr (red squares) to the gating $\tau_{\tau\,fast}$ (black triangles) and the gating $\tau_{slow}$ (open circles) obtained from a double exponential fit to the gating charge ($n = 4$). (**B**) As in A, but for R1C-qBBr:W454A;E247W ($n = 4$).

The online version of this article includes the following source data for figure 9:

**Source data 1.** Source data for fluorescence and gating $\tau$ in Figures 9A and 9B.

charge. However, when the dwell time of the intermediate state is brief, then the fluorescence change is monotonic and has two time constants. In all our studies, we do not see increases and decreases of fluorescence during a single pulse. Instead, fluorescence changes are always in the same direction (either an increase or a decrease), albeit with different kinetics and more than one time constant. The different kinetics observed between different constructs, therefore, may be explained by the brief dwell time in the intermediate state. Additionally, the introduction of a W mutation at different residues of the channel can alter the underlying gating current kinetics. Finally, the fluorescence signals from some constructs are too small (e.g., R2C-qBBr:F416W) to produce useful kinetic information. Consequently, although it would be ideal to be able to directly compare kinetics of one construct to another, the measurement of intensity is the main technical limitation that precludes us from doing so with the present technique. Future improvement of measuring fluorescence lifetime as a function of time, for example, observing changes in fluorescence lifetime during the applied pulse, would allow different construct comparisons.

However, within the same construct, voltage pulses of different durations or potential can be used to interrogate different transitions of the voltage sensor, and the kinetics of different processes from the same construct can then be compared. For example, the deactivation kinetics of the fluorescence change produced by R1C-qBBr:W454A;T326W are more rapid than its activation kinetics, suggesting that the interaction of R1C-qBBr with T326W is different during activation than during deactivation. Additionally, we observe that the QV and FVs of deactivation have a leftward shift to that of activation, thus demonstrating that the movement of individual gating charges follows the leftward shift observed between deactivation and activation.

Another example is the slow R1C-qBBr:W454A;F290W fluorescence change upon hyperpolarization. The kinetics of this interaction are markedly different from the kinetics of the main motion of the voltage sensor. This result gives direct evidence that R1 can populate a region of the sensor closer to the intracellular side when the membrane is strongly hyperpolarized and provides a molecular basis for the Cole–Moore shift.

We can also, within a construct, compare the kinetics of the charge movement (time integral of the gating current) to the kinetics of the fluorescence. For example, R1C-qBBr (*Figure 9A*) and R1C-qBBr:W454A;E247W (*Figure 9B*) display fluorescence curves that are well fit by a single exponential and gating charge kinetics that are well fit by a double exponential. In each case, the fluorescence kinetics align well with the slow component of the gating kinetics, suggesting that R1C-qBBr's rotation from facing toward the VSD to facing toward the pore happens later in the gating process than much of the movement of the gating charge (*Figure 9*).

## Pathways of discrete gating charges

Our data indicate that in the normal resting state, it appears that R1 does not come into close contact with F290 or L294 in the S2 segment, while R2 does come into proximity with L294. In the normal active state, R1 comes into close contact with W454 and Y415, while R2 comes into close contact with Y415, but does not interact with W454. One observation potentially related to the position of the resting state is that two of our tested constructs were lethal: R1C:W454A;I287W and R2C:F290W. Whether the lethality of these constructs stems from a disruption of the normal resting state interactions of discrete gating charges with the gating pore remains to be investigated. However, R1 closely interacting with I287 in the resting state is in good agreement with earlier findings using disulfide bonding (*Campos et al., 2007*), fluorimetry (*Pathak et al., 2007*), and modeling (*Henrion et al., 2012*; *Vargas et al., 2011*).

The ability to optically track a gating charge in real time is a powerful technique, despite its limitations. It allows for the combination of crystallographic studies and functional data to understand the movement of the VSD. In general, our data with R1 and R2 support a sliding helix model of movement of the gating charges but add a more detailed view of the movement. Each discrete gating charge undergoes both a rotation with translation with an angle of about 60° with respect to the plane of the membrane. Interestingly, the rotation of the R1 appears greater than the rotation of the R2, suggesting that some of the movement may be coming from the movement of the side chain, or a transient change of the helical conformation from an α helix to a $3_{10}$ helix (*Bassetto et al., 2020*; *Chakrapani et al., 2010*; *Henrion et al., 2012*; *Schwaiger et al., 2011*) rather than exclusively from the movement of the voltage sensor backbone. We expect that this work can be expanded with molecular modeling. We may be able to test whether the charges move independently of each other, with the first and second gating charge moving first and then tugging the rest of the VSD along via the backbone of the S4 segment (*Horng et al., 2019*).

The qBBr fluorescence data constrain the normal resting state of R1 to a position extracellular to F290 and indicate that interactions between this residue and R1 may be responsible for the Cole–Moore shift by moving the first charge deeper into the intracellular side. This result is consistent with the shallow *QV* curve recorded at very negative potentials and also with the fact that the 'closed state' is in fact a collection of closed states in agreement with the increase in entropy when evolving from the open to the closed state (*Claydon et al., 2007*; *Rodríguez et al., 1998*). These multiple steps of the S4 moving within the closed state have also been suggested by modeling of activation transitions (*Cox et al., 1997*; *Ledwell and Aldrich, 1999*; *Schoppa and Sigworth, 1998*; *Zagotta et al., 1994a*). Importantly, while the F290W mutation causes a leftward shift of the *QV* and *GV* curves (*Lacroix et al., 2011b*; *Monks et al., 1999*; *Tao et al., 2010*), the fluorescence kinetics of R1C-qBBr:W454A;F290W do not match the time course of the R1C-qBBr gating charge (*Figure 7—figure supplement 1*). However, we cannot preclude the possibility that the combination of R1C-qBBr;W454A with F290W could result in a construct that may be functionally distinct from the native protein.

## Extension to other voltage sensors and gating charges

Numerous voltage sensors have been examined with site-directed fluorimetry: Shaker (*Cha and Bezanilla, 1997*; *Mannuzzu et al., 1996*), other Kv1-type channels (*Peters et al., 2009*; *Vaid et al., 2008*), Kv7 (*Kim et al., 2017*; *Osteen et al., 2010*; *Ruscic et al., 2013*), Kv10 and Kv11 (*Schönherr et al., 2002*; *Smith and Yellen, 2002*), Nav (*Cha et al., 1999*; *Chanda and Bezanilla, 2002*; *Wang et al., 2016*), voltage-gated calcium channels (*Pantazis et al., 2014*), BK (*Savalli et al., 2006*), HCN (*Bruening-Wright and Larsson, 2007*; *Dai et al., 2019*), Catsper (*Arima et al., 2018*), and CiVSP (*Kohout et al., 2008*; *Villalba-Galea et al., 2008*). Extension of the qBBr WY-induced quenching technique to other voltage sensors is expected to provide new findings because this technique directly measures the relative displacement between individual charges and amino acid residues residing in their path of movement during voltage sensing. As a proof-of-principle, we demonstrated qBBr can be applied to CiVSP (*Murata et al., 2005*), an evolutionarily distant voltage sensor. Interestingly, we uncovered an interaction between R1 of CiVSP and a tyrosine residue in the S3-S4 linker that is in good agreement with a reported crystal structure of the VSD in the active state (*Li et al., 2014*). While tyrosine-induced quenching of qBBr is weaker than tryptophan-induced quenching, the potential for either amino acid to provide qBBr quenching is potentially useful. Additionally, although any amino acid substitution can change channel structure and function, the feasibility of using tyrosine as a

quencher could help minimize some of the limitations inherent in substituting the bulkier tryptophan residue into the channel. It should also be noted that qBBr optical mapping should also be useful in conducting channels to investigate the movement of discrete gating charges during conductive events and correlate features of gating with ion conductance.

Further mapping of additional voltage sensor pathways with the qBBr method described here should improve our understanding of how voltage sensors and their discrete gating charges move and provide new insight into where the electric field is focused, which side charges interact with the VSD, and the differences in the motions made by unique gating charges. Such qBBr mapping experiments may have additional relevance as recent resting state structures of voltage-sending domains have suggested that there may be a diversity of voltage sensor and gating charge pathways amongst different VSDs (*Gao et al., 2021*; *Lee and MacKinnon, 2019*; *Li et al., 2014*; *She et al., 2018*; *Wisedchaisri et al., 2019*). Finally, a present limitation of the technique is that qBBr mapping of gating charges is limited by labeling accessibility to gating charge residues mutated to cysteine. For example, we were unable to link qBBr to the third most extracellular Shaker gating charge (R368). Future experiments using unnatural amino acid versions of qBBr or other fluorescent amino acids with characteristics similar to qBBr (*Leisle et al., 2015*) may allow for the expansion of this technique to additional charges in the voltage sensor.

# Materials and methods

## Key resources table

| Reagent type (species) or resource | Designation | Source or reference | Identifiers | Additional information |
|---|---|---|---|---|
| Biological sample (*Xenopus laevis*, female) | Oocytes | Nasco | Cat# LM00531 | |
| Strain, strain background (*Escherichia coli*) | XL10-Gold Ultracompetent Cells | Agilent | Cat# 200315 | Competent cells |
| Genetic reagent (*D. melanogaster*) | *Shaker* Δ6–46W434F background template | *Hoshi et al., 1990*; *Perozo et al., 1993* | | |
| Genetic reagent (*Ciona intestinalis*) | *CiVSP* C363S background template | *Murata et al., 2005* | | |
| Commercial assay or kit | mMESSAGE mMACHINE T7 Transcription Kit | ThermoFisher | Cat# AM1344 | |
| Commercial assay or kit | mMESSAGE MACHINE SP6 Transcription Kit | ThermoFisher | Cat# AM1340 | |
| Commercial assay or kit | QuikChange II site directed mutagenesis | Agilent | Cat# 200,523 | |
| Chemical compound, dye | Monobromo(trimethylammonio) bimane (qBBr) | Sigma Aldrich | Cat# 71028; CAS# 71418-45-6 | |
| Chemical compound, dye | Bromotrimethylammoniumbimane bromide (qBBr) | Toronto Research Chemicals | Cat# B688500; CAS# 71418-45-6 | |
| Chemical compound, drug | *N*-Methyl-D-glucamine | Sigma-Aldrich | Cat# M2994; CAS# 6284-40-8 | |
| Chemical compound, drug | Methanesulfonic acid | Sigma-Aldrich | Cat# 471356; CAS# 75-75-2 | |
| Software, algorithm | MATLAB | Mathworks | 2014b or earlier; RRID:SCR_001622 | |
| Software, algorithm | UCSF Chimera | UCSF | 1.10.1; RRID:SCR_004097 | |
| Software, algorithm | Prism | GraphPad | 6; RRID:SCR_002798 | |

## Generation of constructs

Mutations of *Shaker* or *CiVSP* DNA were made on the *Shaker*Δ6–46W434F background (*Hoshi et al., 1990*; *Perozo et al., 1993*) or the *CiVSP* C363S background (*Murata et al., 2005*), respectively, using

the QuikChange II site directed mutagenesis kit (Agilent, Santa Clara, CA) with primers purchased from Integrated DNA Technologies. *Shaker* DNA was linearized using NotI (New England Biolabs, Ipswich, MA) and *CiVSP* DNA was linearized using XbaI (New England Biolabs); both were cleaned up with a NucleoSpin Gel and PCR Clean-up kit (Macherey-Nagel, Bethlehem, PA). *Shaker* and *CiVSP* cRNAs were then synthesized from linearized DNA using the mMESSAGE mMACHINE T7 or SP6 transcription kit, respectively (Life Technologies, Carlsbad, CA).

## Oocyte preparation

Oocytes were harvested from *X. laevis* in accordance with experimental protocols approved by the University of Chicago Animal Care and Use Committee. Unless specified, all chemicals were obtained from Sigma-Aldrich (St. Louis, MO). Following collagenase digestion of the follicular membrane, oocytes were maintained in standard oocyte solution containing 96 mM NaCl, 2 mM KCl, 1 mM $MgCl_2$, 1.8 mM $CaCl_2$, 10 mM HEPES, and 50 µg/ml of gentamicin, set with NaOH to pH 7.4, for up to 36 hr prior to injection with 50 ng of cRNA. Following cRNA injection, oocytes were incubated at 16 °C in standard oocyte solution for 3–6 days prior to recording.

## Labeling with qBBr

The labeling solution consisted of depolarizing solution comprised of 120 mM KCl, 2 mM $CaCl_2$ and 10 mM HEPES at pH 7.4 with 1–2 mM qBBr (Sigma-Aldrich or Toronto Research Chemicals, North York, ON, Canada) added fresh to the solution upon each preparation. Oocytes were maintained in the solution for at least 15 min and removed and washed in standard oocyte solution 5–20 min before recordings on them were performed. At the longest, oocytes were maintained in labeling solution for 120 min; however, due to the positive charge and impermeability of qBBr, no differences were observed between oocytes labeled for long durations versus short durations.

## Simultaneous fluorescence and electrophysiological recordings

Simultaneous electrophysiological and fluorescence recordings were performed at room temperature using the cut-open oocyte voltage-clamp technique, similar to prior descriptions (*Cha and Bezanilla, 1997*; *Lacroix et al., 2012*; *Villalba-Galea et al., 2008*). External solution for gating recordings contained 115 mM *N*-methyl-D-glucamine, 2 mM Ca(OH)$_2$, and 20 mM HEPES taken to a pH between 7.4 and 7.5 with methanesulfonic acid. Internal solution was like external solution, but with 2 mM EGTA in place of Ca(OH)$_2$. For ionic recordings, external solution contained 12 mM K-methanesulfonic acid, 108 mM *N*-methyl-D-glucamine, 10 mM HEPES, 0.1 mM EDTA taken to a pH between 7.4 and 7.5 with methanesulfonic acid. Internal solution contained 120 mM K-methanesulfonic acid, 10 mM HEPES, 2 mM EGTA taken to a pH between 7.4 and 7.5 with methanesulfonic acid. Pipettes were pulled at a resistance of 0.2–1.0 MΩ and filled with 3 M KCl. Excitation was performed with a mounted 420 nm LED (ThorLabs, Newton, NJ) reflected by a 455 nm long-pass dichroic (Chroma, Bellows Falls, VT) through a ×40 water-immersion objective (LUMPlan FL N, Olympus, Center Valley, PA); emission was collected through the dichroic and a 475 nm long-pass filter (Chroma). Emission was integrated over each sampling period through a home-built integrator, collected by a PIN-020A photodiode (UDT Technologies, Torrance, CA), and amplified by a patch-clamp amplifier (L/M-EPC-7, LIST Medical Electronics, Darmstadt, West Germany). Voltage-clamp and electrical measurements were performed with a CA-1B amplifier (Dagan, Minneapolis, MN). The LED and voltage-clamp were controlled through Gpatch, an in-house acquisition program, and an SB6711-A4D4 board (Innovative Integration, Simi Valley, CA). Recordings at each voltage step were an average of four traces, taken consecutively, sampled at 50–100 kHz and filtered at 5–10 kHz.

## Data analysis

Recordings were filtered and analyzed offline using custom MATLAB (Mathworks, Natick, MA) scripts (*Treger et al., 2015*) available as source code. Filtering of fluorescence for display and analysis was performed using a digitized Bessel filter with a cutoff frequency between 500 Hz and 1 kHz. $\Delta F/F_0$ was calculated following a linear baseline subtraction of the period 6–1.5 s prior to the electrical potential change of interest. Charge was calculated from gating currents following linear baseline subtraction of user-defined periods based on when gating current amplitudes returned to zero. Time constant measurements of kinetics of both fluorescence and gating were calculated using either a

single exponential fit or a weighted double exponential fit to the rising or falling phase of the fluorescence as appropriate, or to the decay component of the gating current.

Images of the Shaker and CiVSP proteins with qBBr and tryptophan substitutions were created using UCSF Chimera (*Pettersen et al., 2004*).

### Quantification, statistical analysis, and simulations

Statistical details of experiments can be found in figure legends. *N* represents the number of oocytes, or biological replicates, and data are shown as mean ± standard error of the mean. Calculations were performed in Prism (GraphPad, La Jolla, CA) or with an in-house analysis program (Analysis). Simulations of single molecule and bulk tryptophan quencher–qBBr interactions were performed in MATLAB using a custom script based on the finding that these interactions follow a Stern–Volmer relationship (*Islas and Zagotta, 2006*).

## Additional information

### Funding

| Funder | Grant reference number | Author |
| --- | --- | --- |
| National Institutes of Health | R01-GM030376 | Francisco Bezanilla |
| National Institutes of Health | F31NS081954 | Michael F Priest |

The funders had no role in study design, data collection, and interpretation, or the decision to submit the work for publication.

### Author contributions

Michael F Priest, Elizabeth EL Lee, Conceptualization, Formal analysis, Investigation, Methodology, Validation, Visualization, Writing – original draft, Writing – review and editing; Francisco Bezanilla, Conceptualization, Formal analysis, Funding acquisition, Investigation, Methodology, Project administration, Resources, Software, Supervision, Validation, Visualization, Writing – original draft, Writing – review and editing

### Author ORCIDs

Elizabeth EL Lee http://orcid.org/0000-0001-8071-6786
Francisco Bezanilla http://orcid.org/0000-0002-6663-7931

### Ethics

This study was performed in strict accordance with the recommendations in the Guide for the Care and Use of Laboratory Animals of the National Institutes of Health. All of the animals were handled according to approved Institutional Animal Care and Use Committee (IACUC) protocols of the University of Chicago. The protocol was approved by the Committee on the Ethics of Animal Experiments of the University of Chicago (Permit Number: 71475).

### Decision letter and Author response

Decision letter https://doi.org/10.7554/eLife.58148.sa1
Author response https://doi.org/10.7554/eLife.58148.sa2

## Additional files

### Supplementary files

- Transparent reporting form
- Source code 1. MATLAB scripts for analyzing data and simulating qBBr fluorescence dynamics.

## Data availability

All data generated or analysed during this study are included in the manuscript and supporting files. Actual records have been provided for Figures 2,3,4,5,7,8.

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
