## [Editor Report]

How the voltage sensor moves to produce voltage-dependent opening of ion channels is still an interesting and open question. The authors measured the presumed movement of the voltage sensor of the Shaker potassium channel using the fluorescence quenching of bimane by tryptophan, a method previously used to measure rearrangements in other proteins. To measure the trajectory of the first two voltage sensor charged residues (R1 and R2), these authors modified R1 and R2 cystine mutant channels with a derivative of bimane that has a positive charge similar to the native arginines. They then mutated residues around the sensor to tryptophan and measured the state-dependent quenching with voltage steps. They observed a pattern for the state-dependent quenching that suggests that the charge pathway during activation is a rotation and tilted translation. Furthermore, they suggest that this pathway for activation is distinct from the pathway for deactivation. Finally, they show initial evidence that this approach could also be applied to the voltage-sensitive phosphatase CiVSP.

---

## [Decision Letter]

Thank you for submitting your article "The trajectory of discrete gating charges in a voltage-gated potassium channel" for consideration by *eLife*. Your article has been reviewed by 3 peer reviewers, and the evaluation has been overseen by a Reviewing Editor and Richard Aldrich as the Senior Editor. The reviewers have opted to remain anonymous.

The reviewers have discussed the reviews with one another and the Reviewing Editor has drafted this decision to help you prepare a revised submission.

Summary:

How the voltage sensor moves to produce voltage-dependent opening of ion channels is still an interesting and open question. The authors measured the presumed movement of the S4 of the Shaker potassium channel using the fluorescence quenching of bimane by tryptophan, a method previously used to measure rearrangements in other proteins. To measure the trajectory of the first two S4 charged residues (R1 and R2), these authors modified R1C and R2C mutant channels with a derivative of bimane called qBBr that has a positive charge similar to the native arginines. They then mutated residues around the S4 to tryptophan and measure the state-dependent quenching with voltage steps in *Xenopus oocytes*. They observed a pattern for the state-dependent quenching that suggests that the charge pathway during activation is a rotation and tilted translation. Furthermore, they suggest that this pathway for activation is distinct from the pathway for deactivation, and that further movement of the voltage sensor below -120 mV is responsible for the Cole-Moore shift in the activation kinetics. Finally, they show initial evidence that this approach could also be applied to the voltage-sensitive phosphatase CiVSP.

The findings are interesting, but some more discussion of the limitations of the technique and the assumptions is required. There are many places in the paper where the results are over interpreted (several examples are listed below).

Essential revisions:

1. The use of the term trajectory in the title and elsewhere, suggests the method reveals the pathway for the movement of the S4 charges, instead of just the resting and activated states (and perhaps deep resting state proposed to be responsible for the Cole-Moore effect). In structural terms, a trajectory is defined as an object position (x, y, z) over time. The word "trajectory" must be replaced by something appropriately describing what is actually measured by the signals.

2. A related concern is that the photoinduced electron transfer that is responsible for quenching is not capable of Å resolution measurements of distance or changes in distance as suggested by the authors (see Mansoor et al.). The residues that quench the resting state (pink in Figure 4 and 5) span a huge range of the length of the voltage-sensor domain (18 Å for R1C-qBBr and 21 Å for R2C-qBBr). Could residues that span such a large range all be involved in quenching qBBr at a particular site? If so, then that suggests that the spatial resolution is quite poor and/or unpredictable because it depends on the rotomeric states of the tryptophan and bimane residues. There is no evidence presented for the argument in the discussion that measuring the changes in quenching with time improves the spatial resolution. This must be adequately justified.

3. The reviewers agree that the conclusion that the pathway for activation is distinct from the pathway for deactivation is not well supported and must be eliminated. The evidence for this conclusion seems to be largely the difference in the steady-state QV and FV curves when measured starting from hyperpolarized vs. depolarized voltages (a phenomena referred to as hysteresis). This behavior is seen in many channels and, in general, is just a manifestation of allostery…if the voltage-sensor movement promotes opening, then opening promotes the voltage-sensor movement (shifts activation to the left). It does not have to arise from a different pathway for the movement of the voltage sensor.

4. The reviewers are also concerned that the paper is far from generous with its citations. For example, for the structure of the Shaker potassium channel they cite Chen et al., 2010, a paper that simply reanalyzed Long and Mackinnon's X-ray diffraction data from 2005. They never site Long et al. This is also true for several other missing citations:

Line 42. Larsson et al. 1996.

Line 59. (Long and Mackinnon 2005)

LIne 88. Henrion et al., 2012,

Line 94 Cole and Moore 1960

LIne 170 Perozo et al. 1994, Zagotta et al. 1994

Line 195. (Long and Mackinnon 2005)

Line 211. Elinder et al. 2001

Line 437. Okamura et al. 2005

Line 59. Long..Mackinnon 2007

Line 141. Some reference for exponential dependence of photoinduced electron transfer

Line 204. Chen et al., 2010

Line 218. Some citations for "crystallographic and functional data"

LIne 340. Zagotta et al., 1994

LIne 577. Dai et al., 2019

The paper should also include some discussion of the S4 moving in multiple steps, with ciations to Zagotta et al., 1994, Schoppa and Sigworth, 1998, Cui and colleagues and others.

It should also include some discusion of cryo-EM structures of the down state of voltage-dependent potassium channels and citations to Jiang (TPC1), Catterall (NavAb), and Mackinnon (HCN).

5. For the quenching/dequenching effects, confidence intervals or statistical comparisons must be provided.

[Editors' note: further revisions were suggested prior to acceptance, as described below.]

Thank you for resubmitting your work entitled "Tracking the movement of discrete gating charges in a voltage-gated potassium channel" for further consideration by *eLife*. Your revised article has been evaluated by Richard Aldrich (Senior Editor) and a Reviewing Editor.

The manuscript has been improved but there are some remaining issues that need to be addressed, as outlined below:

Although the reviewers are mostly satisfied with the revisions, they remain concerned about overinterpretation of the results. An additional and rather straightforward revision is required to address their concerns as described in their comments and to "tone down" some of the more emphatic statements about the interpretations. Speculation is fine, but it should be identified as such.

*Reviewer #1:*

The authors have responded well to my comments

*Reviewer #2:*

The revisions partially address the original concerns. However, one main concern was that PET between qBBr and tryptophan is highly dependent on the rotomeric state of the amino acid and does not accurately reflect backbone distances. This concern was not totally alleviated. The manuscript still makes numerous references to distances and movements that are not well determined by the data. The proposed movement "that upon activation both R1 and R2 undergo a tilted translation from the intracellular to the extracellular side of the membrane, together with a rotation" seems like an overinterpretation. In combination with large and not fully characterized functional effects of some of the mutants, uncertainty in the degree of labeling, and variability in the background fluorescence, the inability to measure distances ultimately limits how definitive the conclusions can be.

*Reviewer #3:*

Overall, the authors have addressed all my previous comments. I have only one remaining suggestion. While the data presented here show that the bimane in R1 does not get close to the W at position 290 in non-hyperpolarized resting states, the conclusion that this is also the case for the native F290 and R1 should be accompanied by a cautionary note. The F290W mutation is well known to cause a significant shift of the channel GV curve to more hyperpolarized potentials (e.g., Monks et al. JGP 1999, Tao et al. Science 2010) as it energetically favors the open state relative to the closed state. The large effect is likely the result of placing a bulky W in what is considered the narrowest part of the "hydrophobic plug" or "charge transfer center". And this is just with the regular-size R1. The larger bimane may have additional effects.

---

## [Author Response]

Summary:How the voltage sensor moves to produce voltage-dependent opening of ion channels is still an interesting and open question. The authors measured the presumed movement of the S4 of the Shaker potassium channel using the fluorescence quenching of bimane by tryptophan, a method previously used to measure rearrangements in other proteins. To measure the trajectory of the first two S4 charged residues (R1 and R2), these authors modified R1C and R2C mutant channels with a derivative of bimane called qBBr that has a positive charge similar to the native arginines. They then mutated residues around the S4 to tryptophan and measure the state-dependent quenching with voltage steps in *Xenopus oocytes*. They observed a pattern for the state-dependent quenching that suggests that the charge pathway during activation is a rotation and tilted translation. Furthermore, they suggest that this pathway for activation is distinct from the pathway for deactivation, and that further movement of the voltage sensor below -120 mV is responsible for the Cole-Moore shift in the activation kinetics. Finally, they show initial evidence that this approach could also be applied to the voltage-sensitive phosphatase CiVSP.The findings are interesting, but some more discussion of the limitations of the technique and the assumptions is required. There are many places in the paper where the results are over interpreted (several examples are listed below).

We thank the reviewers for their comments. We have significantly changed our discussion of the results in light of the comments below, and added data as requested in Figure 1—figure supplement 1, Figure 2, Figure 2—figure supplement 1, and Figure 5—figure supplement 1, Figure 9, and Table 1.

Essential revisions:1. The use of the term trajectory in the title and elsewhere, suggests the method reveals the pathway for the movement of the S4 charges, instead of just the resting and activated states (and perhaps deep resting state proposed to be responsible for the Cole-Moore effect). In structural terms, a trajectory is defined as an object position (x, y, z) over time. The word "trajectory" must be replaced by something appropriately describing what is actually measured by the signals.

We have kept the word ‘trajectory’ in the section on the underlying theory of qBBr tracking, as in this idealized conception of the technique, this word would be applicable. It has been replaced in every other instance, including in the title. Generally, these replacements have described ‘tracking the movement’ and have been put in blue font color in the text.

2. A related concern is that the photoinduced electron transfer that is responsible for quenching is not capable of Å resolution measurements of distance or changes in distance as suggested by the authors (see Mansoor et al.). The residues that quench the resting state (pink in Figure 4 and 5) span a huge range of the length of the voltage-sensor domain (18 Å for R1C-qBBr and 21 Å for R2C-qBBr). Could residues that span such a large range all be involved in quenching qBBr at a particular site? If so, then that suggests that the spatial resolution is quite poor and/or unpredictable because it depends on the rotomeric states of the tryptophan and bimane residues. There is no evidence presented for the argument in the discussion that measuring the changes in quenching with time improves the spatial resolution. This must be adequately justified.

The reviewers raise numerous good points in this comment.

We agree with the reviewers that the quenching effect is not capable of Å resolution measurements. We do not believe we ever suggested it was capable of such spatial resolution, and are sorry if we unintentionally did so. We have slightly reworded our previous reference to Mansoor et al., 2004 to make it clearer that the maximum quenching distance of the technique is between 10 and 15 angstroms.

Regarding the long span of residues involved in quenching qBBr, we have clarified that, as described by our simulation, “our data do not show that in the resting state, qBBr is in close contact with each residue it unquenches from upon activation. Indeed, the movement of qBBr past a tryptophan will produce an unquenching effect as long as its final distance from that tryptophan is greater than its initial distance from that tryptophan (Figure 1C, Figure 1—figure supplement 1).” We also specify that due to the presence of rotameric states of the side chains and the variability in endogenous background fluorescence of the oocytes that absolute calibration of distance using this method is not possible, and explain that spatial resolution is improved by “increasing the number of substituted quenching residues found to interact or not interact with the gating charge.”

Finally, we have removed the section in the discussion arguing that the spatial resolution is improved by measurement of a dynamic signal.

3. The reviewers agree that the conclusion that the pathway for activation is distinct from the pathway for deactivation is not well supported and must be eliminated. The evidence for this conclusion seems to be largely the difference in the steady-state QV and FV curves when measured starting from hyperpolarized vs. depolarized voltages (a phenomena referred to as hysteresis). This behavior is seen in many channels and, in general, is just a manifestation of allostery…if the voltage-sensor movement promotes opening, then opening promotes the voltage-sensor movement (shifts activation to the left). It does not have to arise from a different pathway for the movement of the voltage sensor.

Thank you. We have removed this conclusion and have altered this section significantly, refocusing on the finding that deactivation transitions in general are also followed by qBBr. We have also excised or substantially altered related text in the abstract and discussion. We also now discuss some of the noteworthy discrepancies between QVs and FVs of individual constructs.

“Specifically, deactivation currents are slowed down in a voltage-dependent manner which corresponds with pore opening (Labro et al., 2012; Lacroix et al., 2011; McCormack et al., 1994; Perozo et al., 1993b). […] Thus, overall, we have strong evidence that the qBBr fluorescence follows deactivation transitions as well as activation transitions, and that qBBr follows the different energetic paths experienced by the voltage sensor during deactivation and activation.”

4. The reviewers are also concerned that the paper is far from generous with its citations. For example, for the structure of the Shaker potassium channel they cite Chen et al., 2010, a paper that simply reanalyzed Long and Mackinnon's X-ray diffraction data from 2005. They never site Long et al. This is also true for several other missing citations:Line 42. Larsson et al. 1996.Line 59. (Long and Mackinnon 2005)Line 88. Henrion et al., 2012,Line 94 Cole and Moore 1960

Done.

Line 170 Perozo et al. 1994, Zagotta et al. 1994

(cited Perozo et al. 1993; Hoshi et al., 1990)

Line 195. (Long and Mackinnon 2005)Line 211. Elinder et al. 2001

Done.

Line 437. Okamura et al. 2005

Murata et al., & Okamura, 2005, cited at the end of the introduction where we first discuss CiVSP.

Line 59. Long..Mackinnon 2007

We have not included this citation, as it references the chimeric potassium channel crystal structure, which was not directly related to this work because it does not have R1.

Line 141. Some reference for exponential dependence of photoinduced electron transfer

We have cited Islas and Zagotta, 2006, and corrected this section of the work to reflect the correct relationship between qBBr and quencher photoinduced electron transfer

Line 204. Chen et al., 2010Line 218. Some citations for "crystallographic and functional data"Line 340. Zagotta et al., 1994Line 577. Dai et al., 2019

Done.

The paper should also include some discussion of the S4 moving in multiple steps, with ciations to Zagotta et al., 1994, Schoppa and Sigworth, 1998, Cui and colleagues and others.

Added requested discussion to Discussion section on ‘Pathways of discrete gating charges’. “These multiple steps of the S4 moving within the closed state have also been suggested by modeling of activation transitions” with appropriate citations. (Pg. 35-36)

It should also include some discusion of cryo-EM structures of the down state of voltage-dependent potassium channels and citations to Jiang (TPC1), Catterall (NavAb), and Mackinnon (HCN).

Thank you. We have included some discussion of these structures in the last paragraph.

“Such qBBr mapping experiments may have additional relevance as recent resting state structures of voltage-sensing domains have suggested that there may be a diversity of voltage sensor and gating charge pathways amongst different voltage-sensing domains (Gao et al., 2021; Lee and MacKinnon, 2019; Li et al., 2014; She et al., 2018; Wisedchaisri et al., 2019).”

5. For the quenching/dequenching effects, confidence intervals or statistical comparisons must be provided.

We have provided a statistical table of the ΔF/F0 of each tryptophan mutant, in Table 1. Thank you.

[Editors' note: further revisions were suggested prior to acceptance, as described below.]

Although the reviewers are mostly satisfied with the revisions, they remain concerned about overinterpretation of the results. An additional and rather straightforward revision is required to address their concerns as described in their comments and to "tone down" some of the more emphatic statements about the interpretations. Speculation is fine, but it should be identified as such.Reviewer #2:The revisions partially address the original concerns. However, one main concern was that PET between qBBr and tryptophan is highly dependent on the rotomeric state of the amino acid and does not accurately reflect backbone distances. This concern was not totally alleviated. The manuscript still makes numerous references to distances and movements that are not well determined by the data. The proposed movement "that upon activation both R1 and R2 undergo a tilted translation from the intracellular to the extracellular side of the membrane, together with a rotation" seems like an overinterpretation. In combination with large and not fully characterized functional effects of some of the mutants, uncertainty in the degree of labeling, and variability in the background fluorescence, the inability to measure distances ultimately limits how definitive the conclusions can be.

Thank you for the opportunity to clarify. When we do mention distances, it is typically with respect to literature references, or indicating sizes of molecules or measuring lengths within structures, or while describing our simulated models. On a final occasion, we use distances to explicitly state that we cannot conclude that the S4 backbone is moving a particular distance (see paragraph in Line 462).

Regarding our descriptions of movements, we would like to note that while rotameric interactions can alter qBBr quenching, the use of multiple constructs alleviates (but does not negate) this concern. “The dependence of qBBr quenching on the rotameric orientation of the dye and the quencher underscores the importance of recording fluorescence changes using multiple constructs with tryptophan substituted at distinct sites.”.

Finally, at the reviewers request, we have updated our characterization of the movements to emphasize that these findings are not a wholly definitive description of the gating charge movement.

“Together these findings are consistent with a pathway for R1 that consists of both an intracellular to extracellular translation, tilted by about 30° with respect to the normal of the membrane plane, and a rotation (Figure 4B).”(Line 260)

“Our fluorescence data from both R1C-qBBr and R2C-qBBr are consistent with a model in which upon activation both R1 and R2 undergo a tilted translation from the intracellular to the extracellular side of the membrane, together with a rotation.” (Line 294)

Reviewer #3:Overall, the authors have addressed all my previous comments. I have only one remaining suggestion. While the data presented here show that the bimane in R1 does not get close to the W at position 290 in non-hyperpolarized resting states, the conclusion that this is also the case for the native F290 and R1 should be accompanied by a cautionary note. The F290W mutation is well known to cause a significant shift of the channel GV curve to more hyperpolarized potentials (e.g., Monks et al. JGP 1999, Tao et al. Science 2010) as it energetically favors the open state relative to the closed state. The large effect is likely the result of placing a bulky W in what is considered the narrowest part of the "hydrophobic plug" or "charge transfer center". And this is just with the regular-size R1. The larger bimane may have additional effects.

We have updated the manuscript to mention this additional concern and references. ”Importantly, while the F290W mutation causes a leftward shift of the QV and GV curves. (Lacroix and Bezanilla, 2011; Monks et al., 1999; Tao et al., 2010), the fluorescence kinetics of R1C-qBBr:W454A;F290W do not match the time course of the R1C-qBBr gating charge (Figure 7—figure supplement 1). However, we cannot preclude the possibility that the combination of R1C-qBBr;W454A with F290W could result in a construct that may be functionally distinct from the native protein.”.